# Selectivity descriptors for the direct hydrogenation of CO₂ to hydrocarbons during zeolite-mediated bifunctional catalysis

Adrian Ramirez [1,4], Xuan Gong[2,4], Mustafa Caglayan [1], Stefan-Adrian F. Nastase [1], Edy Abou-Hamad[3], Lieven Gevers[1], Luigi Cavallo [1], Abhishek Dutta Chowdhury [2✉] & Jorge Gascon [1✉]

Cascade processes are gaining momentum in heterogeneous catalysis. The combination of several catalytic solids within one reactor has shown great promise for the one-step valorization of C1-feedstocks. The combination of metal-based catalysts and zeolites in the gas phase hydrogenation of CO₂ leads to a large degree of product selectivity control, defined mainly by zeolites. However, a great deal of mechanistic understanding remains unclear: metal-based catalysts usually lead to complex product compositions that may result in unexpected zeolite reactivity. Here we present an in-depth multivariate analysis of the chemistry involved in eight different zeolite topologies when combined with a highly active Fe-based catalyst in the hydrogenation of CO₂ to olefins, aromatics, and paraffins. Solid-state NMR spectroscopy and computational analysis demonstrate that the hybrid nature of the active zeolite catalyst and its preferred CO₂-derived reaction intermediates (CO/ester/ketone/hydrocarbons, i.e., inorganic-organic supramolecular reactive centers), along with 10 MR-zeolite topology, act as *descriptors* governing the ultimate product selectivity.

[1] KAUST Catalysis Center (KCC), King Abdullah University of Science and Technology (KAUST), Thuwal 23955, Saudi Arabia. [2] The Institute for Advanced Studies (IAS), Wuhan University, Wuhan 430072 Hubei, P. R. China. [3] Imaging and Characterization Department, KAUST Core Labs, King Abdullah University of Science and Technology (KAUST), Thuwal 23955, Saudi Arabia. [4] These authors contributed equally: Adrian Ramirez, Xuan Gong. ✉email: abhishek@whu.edu.cn; jorge.gascon@kaust.edu.sa

Carbon capture and utilization (CCU) is critical to mitigating global warming[1,2]. The success of the CCU approach relies heavily upon the amount of $CO_2$ that can be stored in the final products[3]. In this spirit, the transformation of $CO_2$ to high-value ($C_{2+}$) hydrocarbons[3], using green hydrogen and renewable electricity[4,5], has become a worldwide research priority (Fig. 1)[4–12]. By producing carbon-rich, high volumetric energy density hydrocarbons, the CCU strategy would allow us to severely reduce $CO_2$ emissions and close the carbon cycle[3].

In this context, thermally catalyzed approaches that rely on bi/multifunctional catalysts (comprising a "redox" metallic catalyst and an "acidic" zeolite) have gained a great deal of attention over the last few years[6,7,12]. The main advantage of such a cascade approach resides in the scope of products: standalone metal catalysts for the hydrogenation of $CO_2$ produce either C1 products at relatively low conversions per pass (e.g., thermodynamically limited methanol synthesis) or low-value C1 products (i.e., methane) or olefinic/paraffinic hydrocarbon mixtures,[11–15] which require further processing and are usually accompanied by undesired products (i.e., $CO/CH_4$)[3,6]. These complex mixtures may, however, be transformed into more interesting products (such as aromatics) upon the addition of a second catalyst, opening the door to direct synthesis of petrochemicals not accessible through standalone metal catalysis[16].

Two main approaches are followed in cascade systems for the hydrogenation of $CO_2$: the transformation of $CO_2$ (i) via reverse water gas shift (RWGS: $CO_2 + H_2 \rightleftharpoons CO + H_2O$) plus Fischer–Tropsch synthesis (FTS) on the metal catalysis, followed by oligomerization/cracking/aromatization reactions on the zeolitic component[17–21], and (ii) using a methanol synthesis catalyst ($CO_2 + 3H_2 \rightarrow CH_3OH + H_2O$)[22–26], in combination with the classical methanol-to-hydrocarbon (MTH)[27] reaction over the zeolite framework. An important issue related to cascade systems is the high selectivity of undesired CO (often more than half of the total products when the second approach based on methanol is followed). This leads to researchers unfairly excluding CO selectivity data and, thus, portrays an unrealistic catalytic profile[6]. To address this issue, we have recently developed a unique catalyst combination comprising potassium superoxide-doped iron oxide ($Fe_2O_3@KO_2$) and an acidic zeolite, which results in low selectivities for undesired CO/ $CH_4$[14,17–19]. The catalytic profile of our standalone $Fe_2O_3@KO_2$ catalyst was in the order of commercial FTS materials[14], where the ultimate selectivity could further be tuned toward desired hydrocarbons (olefins/aromatics) through variation of the zeolite components only[17–19]. However, to arrive at further performance improvements, it is necessary to better understand the chemistry at play, especially on the zeolite components whose interplay in the overall cascade reaction mechanism is barely touched on the state of the art[12].

To broaden the scope and establish a new state of the art for this CCU approach (Fig. 1), we have studied eight different zeolites (ZSM-5, MOR, SAPO-34, ZSM-22, FER, BETA, ZSM-58, and Y) in combination with the $Fe_2O_3@KO_2$ catalyst. We find that the different zeolite topologies can be classified into four distinct groups in terms of selectivity: (i) light olefins (MOR, SAPO-34, ZSM-58, BETA, Y), (ii) paraffins (FER), (iii) long (olefinic) hydrocarbons (ZSM-22), and (iv) aromatics (ZMS-5)[3]. To trace the origin of such selectivity differences and to unravel complex reaction mechanisms, in-depth advanced magic angle spinning (MAS) solid-state nuclear magnetic resonance (ssNMR) spectroscopy has been performed on the postreacted zeolite materials. Furthermore, using fully $^{13}C$ isotope-enriched $CO_2/CO$ in the reactant feed increased the sensitivity by >99% (cf. $^{13}C$-natural abundance: ~1.1%), which allowed us to perform multidimensional ssNMR experiments to decode the structure of zeolite-trapped organics and gain insight into the reactivity of the different frameworks[28–31]. To support our ssNMR and catalytic experiments, computational calculations were performed aiming at a finer understanding regarding the stability and involvement in the reaction cycle of the organic-carbonylated species in the zeolite phase. According to our results, selectivity patterns are primarily driven by the extent of consumption of RWGS-derived CO over the zeolite and the formation of carbonylated species (specifically ketenes and its derived ester or (di-)ketone), along with typical hydrocarbons (olefinic/aromatics/paraffins). In a similar fashion as in the MTH process[27,31–33], we validate that the in situ formed hybrid inorganic–organic material or supramolecular reactive centers[34–36], composed by the inorganic zeolite acid sites and surrounding lattice and its trapped organic compounds —hydrocarbon pool (HCP), is the active catalyst and that reactivity is finally defined by both the zeolite framework topologies and the trapped organics.

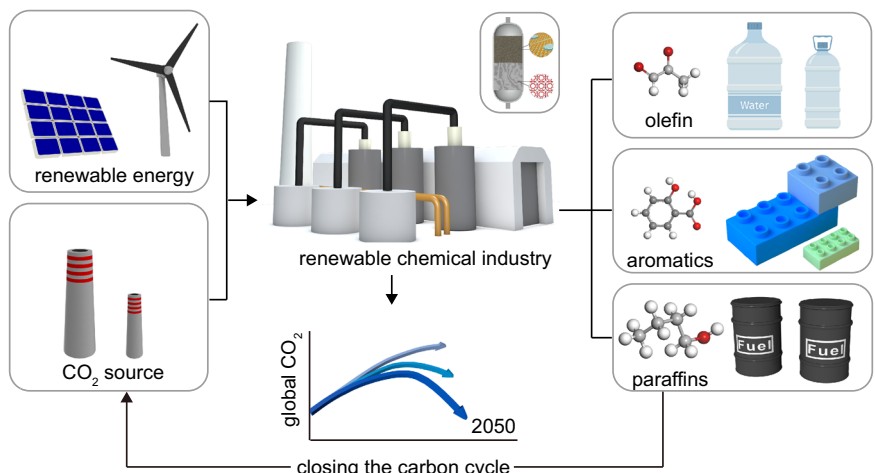

**Fig. 1 The big picture.** Exploring the mitigating potential of carbon capture and utilization (CCU) on climate change, through decoupling chemical production from fossil resources. The potential success of this CCU-based approach heavily relies upon the amount of $CO_2$ stored in the final products. Herein this work, through the production of carbon-rich high volumetric energy density ($C_{2+}$) chemicals (olefins, aromatics, and paraffins) from $CO_2$, our CCU strategy would allow us to "bend the curve" of global $CO_2$ emission from "cradle to gate" in the chemical industry, and thus, to take the right step toward a closed-loop anthropogenic carbon cycle.

## Results

To illuminate the effect of zeolite phase on the overall reaction process, it is mandatory to alter the zeolite alone from the $Fe_2O_3@KO_2$/zeolite-based material, without changing the standalone metal catalyst (see Supplementary Methods and Discussion). An in-depth characterization of standalone $Fe_2O_3@KO_2$ catalyst has been earlier reported by us[14,17–19], while we refer to Supplementary Information for the additional characterization data on the metallic phase (on both fresh and spent $Fe_2O_3@KO_2$ catalysts) by using Raman (micro)spectroscopy and air-protected capillary single-crystal X-ray diffraction (cf. Supplementary Discussion) as well as the fundamental characterization of zeolites (see Supplementary Tables 1 and 2, Supplementary Figs. 1 and 2, and Supplementary Discussion). These frameworks were selected based on their topological features: four 3D pore networks (ZSM-5, SAPO-34, BETA, and Y), three 2D structures (MOR, FER, and ZSM-58), and one 1D pore (ZSM-22) systems.

**Catalysis data.** The multifunctional $Fe_2O_3@KO_2$/zeolite system was assembled by combining the metallic part with zeolites in a dual-bed configuration with a mass ratio of 1:1 (Fig. 2, also see Supplementary Methods for experimental details). Under the studied conditions (30 bar, 375 °C, $H_2/CO_2 = 3$, and $10,000\ mL \cdot g^{-1} \cdot h^{-1}$), the standalone iron catalyst (first column in the left, Fig. 2a) led to a $CO_2$ conversion of 47% with a selectivity toward light olefins, ($C_2$–$C_9$) paraffin, long-chain ($C_5$=–$C_9$=) olefin, larger hydrocarbons, and total CO selectivity of 38%, 9%, 30%, 6%, and 17%, respectively[14]. When a zeolite is combined with the metallic catalyst, CO selectivity is reduced, reaching its minimum value with BETA zeolite. $CO_2$ conversion remained unchanged in all cases, showing that, unlike CO, none of these zeolites can activate $CO_2$[17,19]. Hydrocarbon distributions in Fig. 2a further illustrate that the light olefin fraction (yellow bars) was slightly increased in most zeolites (MOR, SAPO, ZSM-58, BEA, Y), while ZSM-22, FER, and ZSM-5 enhanced the formation of longer olefins (green bars), paraffins (blue bar), and aromatics (purple bar), respectively. To further emphasize the influence of the zeolite on hydrocarbon distributions, Fig. 2b highlights the selectivity changes on each hydrocarbon components for each zeolite (with respect to the standalone $Fe_2O_3@KO_2$ catalyst). The consumption of CO upon the introduction of zeolite could easily be rationalized, as CO selectivity (red bars) is decreased ca. 15%, which indicates the (co-)existence of multiple carbonylated species as reactive intermediates. Similarly, the lighter olefins selectivity is increased by ~15% for most zeolites (Fig. 2b). Surprisingly, only 10 MR zeolites did not follow this trend: ZSM-22, FER, and ZSM-5. Among them, ZSM-22 consumed CO and lighter olefins to yield higher olefins and paraffins, clearly advocating for C–C coupling reactions. Similarly, FER converted light/heavy olefins into paraffins (~128% increase), while ZSM-5 transformed both CO and olefins into aromatics and paraffins.

Given these results, the different zeolites can be classified into four distinct groups: those that incorporate CO to form (i) light olefins (MOR/SAPO-34/ZSM-58/BETA/Y), (ii) long (olefinic) hydrocarbons (ZSM-22), and those where hydrogen transfer is dominant and results in the formation of (iii) paraffins (FER) or (iv) aromatics (ZMS-5). As a representative of these four groups, we have selected MOR, ZSM-22, FER, and ZSM-5 for further mechanistic investigations. Figure 3 displays a detailed hydrocarbon distribution along with a direct comparison to the standalone $Fe_2O_3@KO_2$ catalyst.

**Solid-state NMR spectroscopy.** Next, ssNMR spectroscopy has been performed on the postreacted zeolite materials after 2 days under reaction conditions (30 bar, 375 °C, $H_2/CO_2 = 3$) using $^{13}C$

isotope-enriched $CO_2$ (see Figs. 4 and 5, Supplementary Figs. 3–17, and additional Supplementary Discussion on ssNMR). It is worth highlighting that experiments using $^{13}CO_2$ in the literature rely on atmospheric pressure that are far from real operational conditions. A complementary set of ssNMR magnetization transfer techniques has been employed to elucidate the molecular structure of the zeolite-trapped organic species based on their mobility. Strategically, by applying either "scalar" through-bond (cf. $^1H$-$^{13}C$ insensitive nuclei enhanced by polarization transfer, INEPT)[37] or "dipolar" through-space (cf. $^1H$-$^{13}C$ cross polarization, CP)[38] magnetization transfer schemes, both mobile (i.e., species with fast tumbling/rotation around the C–C axis or locally mobile groups) and rigid/limited-mobile (i.e., species physisorbed in/on the zeolite) organics have been distinguished, respectively[28–31]. Furthermore, direct excitation (DE) has also been applied to detect all chemical species, including species exhibiting intermediate dynamics. This strategy, although conceptualized to spectrally resolve biomolecules with a high or restricted mobility[39], has recently been successfully applied to elucidate accurate reaction mechanisms in zeolite catalysis[28–31]. In the 1D $^1H$-$^{13}C$ CP, $^1H$-$^{13}C$ INEPT, and $^{13}C$ DE ssNMR spectra of the postreacted zeolites (see Supplementary Fig. 3), the following three features were primarily observed: (i) 5–40 ppm aliphatic, (ii) 110–150 ppm (methylated)aromatic/olefinic, and (iii) 180–220 ppm carbonyl moieties[28–31,40,41]. Upon adopting different magnetization transfer schemes, the nonidentical intensity profile on different samples signifies the influence of "mobility-dependent" host–guest chemistry.

ssNMR experiments targeted to mobile species revealed the existence of paraffins and carbonylated moieties but the absence of unsaturated "C=C bonds" typically found in aromatics and olefins (Fig. 4 and Supplementary Figs. 6–11). Aliphatic regions are overwhelmed with the presence of methyl (–$CH_3$), methylene (–$CH_2$–), methine (>CH–), and quaternary carbon (>C<) groups, indicative of both linear and branched paraffin-based species (Fig. 4). For example, in $^{13}C$-$^1H$ correlation spectrum in ZSM-5 (Fig. 4a, b), ethane (Δ: 8.0 ($^{13}C$)/~1.02 ($^1H$) ppm) and isobutane (▪: 26.1 ($^{13}C$)/~1.43($^1H$) ppm), typical hydrogen transfer products from shorter olefins, were readily distinguishable[30,31]. $^1H$ resonances of these species are again relatively broader than usual, advocating for the heterogeneity of the local environment within zeolite, i.e., a species resides in a different local environment[29]. While combining $^{13}C$-$^{13}C$ and $^{13}C$-$^1H$ correlation experiments in ZSM-22 (Fig. 4a, b), additionally, n-butane could be identified as well (○: 24.1 ($^{13}C$)/2.63($^1H$) ppm ↔ 14.5 ($^{13}C$)/1.02 ($^1H$) ppm). Quaternary carbon groups in $^{13}C$-$^{13}C$ DE experiments (35–45 ppm, Fig. 4a), which do not have any corresponding $^1H$ resonances (Fig. 4b, d), could not be assigned to any particular structure due to their substituent's spectral crowding. In MOR, at least $C_4$-butane could be identified (○: 24.4 ($^{13}C$)/~1.38 ($^1H$) ppm ↔ 14.3 ($^{13}C$)/1.55 ($^1H$) ppm), where the broader scalar-based $^1H$-$^{13}C$ correlations imply rigidity of MOR-trapped species (Fig. 4c, d). In FER, a cross peak between 26.1 ($^{13}C$)/1.02 ($^1H$) ppm and 30.3 ($^{13}C$)/2.27 ($^1H$) ppm in $^{13}C$-$^{13}C$ correlations highlights the presence of (>$C_4$)-long-chain alkanes (□: Fig. 4b, c). Next, in the carbonyl region, a diverse set of resonances have been detected (excluding MOR, see Fig. 4e and Supplementary Figs. 12 and 13). In FER, two organic carbonyl signals at 176.6 and 211.9 ppm exhibited cross peaks with a $^{13}C$-methyl signal at 19.2 and 29.8 ppm, respectively, which we attributed to acetate group on zeolite Brønsted acid site (BAS) (Θ: –Si-O(COCH$_3$)-Al) and acetone (Ω), respectively (Fig. 4e, f)[28,31,40–43]. The zeolite acetate has also been detected on ZSM-22 (Θ: 180.2/18.8 ppm, Fig. 4e). An interesting common carbonyl-based cross peak in both ZSM-5 (∂: 197.5/23.05 ppm)

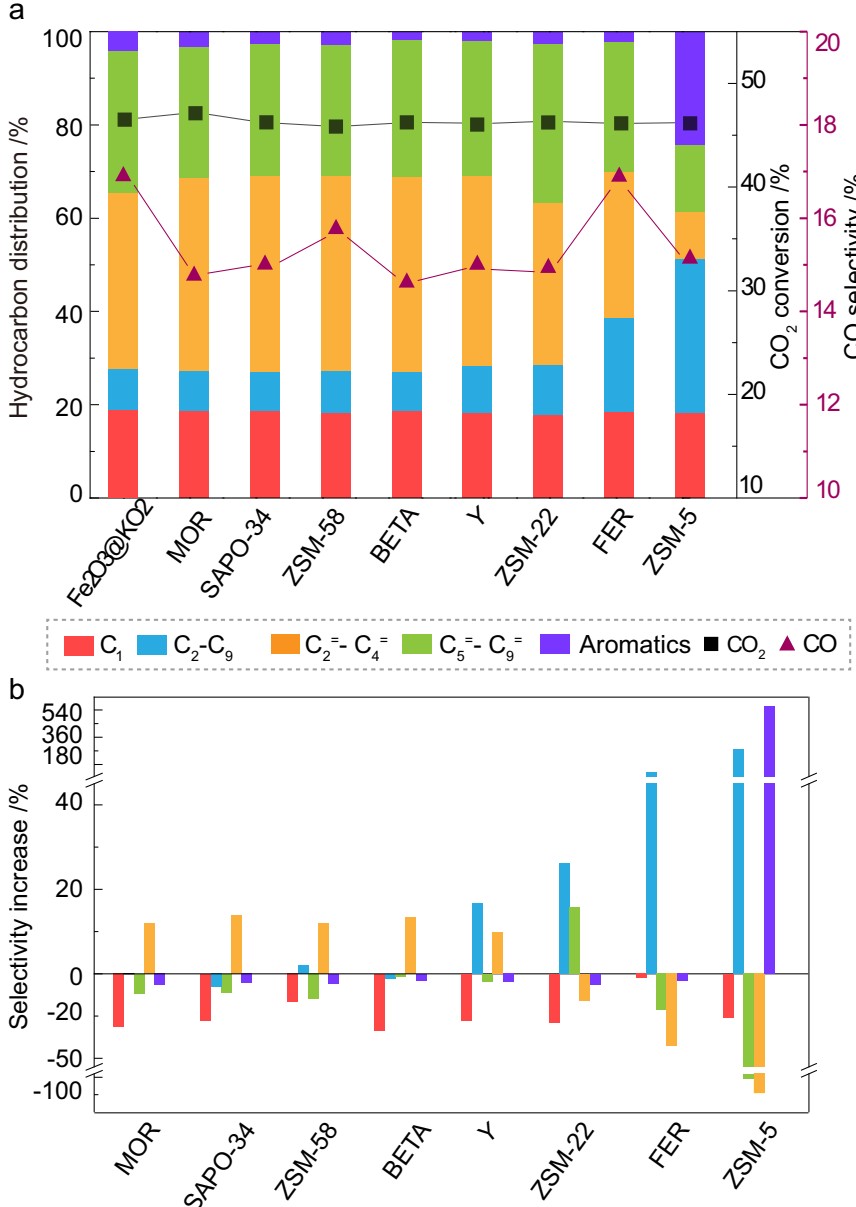

**Fig. 2 Catalytic performance of the Fe₂O₃@KO₂/zeolite bifunctional material catalyzed hydrogenation of CO₂ process. a** Comparison of catalytic data using different zeolites. **b** Detailed illustration of the (net) selectivity increase (or decrease) of individual hydrocarbon groups upon introducing zeolites with respect to the standalone Fe/K catalyst. Reaction condition: 30 bar, 375 °C, $H_2/CO_2 = 3$, and 10,000 mL·g⁻¹·h⁻¹ at a time on stream of 48 h.

and ZSM-22 (∂: 198.9/24.12 ppm) ascribed to diacetyl (CH₃CO-COCH₃, Fig. 4e, f). The simultaneous existence of acetone, acetate, and diacetyl groups strongly suggests the involvement of ketene (CH₂CO) as an active reaction intermediate (vide infra, see Supplementary Fig. 24)[16,31,44–48].

While probing rigid molecules (Fig. 5 and Supplementary Figs. 4 and 5), unsaturated (olefins/aromatics) and (branched) saturated hydrocarbons were predominantly distinguishable (i.e., hydrogen-transferred species). Expectedly, paraffins particularly dominated in zeolites ZSM-5 and ZSM-22, where the presence of tetramethylethane (○: 32.5 (¹³C)/2.25 (¹H) ppm ↔ 23.9 (¹³C)/1.37 (¹H) ppm, Fig. 5a, b) could be identified[30,31]. Although a few other correlations (◊: 36.5 (¹³C)/~2–3 (¹H) ppm ↔ 11.3 (¹³C)/1.37 (¹H) ppm, 45.6 (¹³C)/~2–3 (¹H) ppm, Fig. 5a, b), unfortunately, could not be unambiguously assigned to any particular structure, this correlation still confirms the existence of branched alkylated paraffinic backbones. Next, methylated

olefins/aromatics were identified in all zeolites (except FER). For example, one methyl resonance at 19.0 ppm (¹³C)/2.25 ppm (¹H) showed direct cross peaks with at least two ¹³C_sp2 resonances, e.g., 128.6 (□) ppm in ZSM-5 and 119.4 (■) ppm in MOR (Fig. 5b, c), where both were further correlated to multiple other carbons in the ¹³C-¹³C correlation spectra (e.g., □: 128.6↔139.1↔146.1 ppm, ■: 119.4↔112.2↔145.9 ppm)[29]. This is a typical signature of methylated (poly)aromatics and long-chain olefins. Interestingly, the overall line width of the ¹³C-¹H HETCOR spectra is exceptionally broad (Fig. 5e), which also sometimes led to more than one peak for the same resonance, implying local heterogeneity[29]. However, FER-trapped species are not quite well resolved in CP-based experiments due to their peculiar high mobility features. Hence, the extent of trapped sp² hydrocarbons is seemingly negligible in FER, but more in MOR, consistent with the previous observation in scalar-based measurements. Another striking feature is the presence of methanol (x:

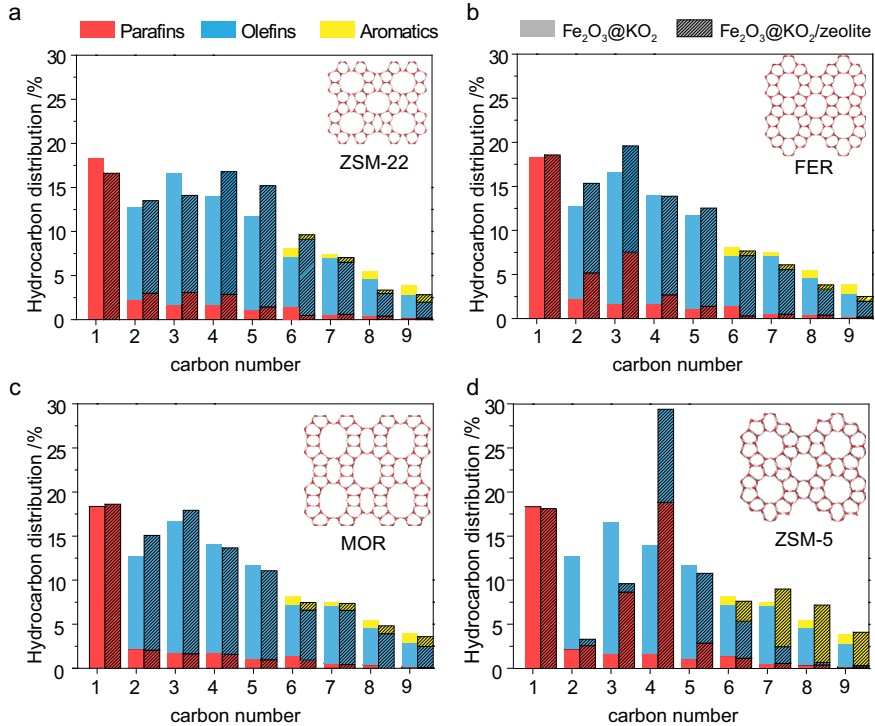

**Fig. 3 Detailed hydrocarbon distribution for selected zeolite frameworks in a dual-bed configuration during Fe$_2$O$_3$@KO$_2$/zeolite bifunctional material catalyzed hydrogenation of CO$_2$ process. a** Fe$_2$O$_3$@KO$_2$/ZSM-22. **b** Fe$_2$O$_3$@KO$_2$/FER. **c** Fe$_2$O$_3$@KO$_2$/MOR. **d** Fe$_2$O$_3$@KO$_2$/ZSM-5. Reaction condition: 30 bar, 375 °C, H$_2$/CO$_2$ = 3, and 10,000 mL·g$^{-1}$·h$^{-1}$ at a time on stream of 48 h.

52.1 ($^{13}$C)/3.24 ($^1$H) ppm) and dimethyl ether (DME, Δ: 63.9 ($^{13}$C)/4.65 ($^1$H) ppm) in ZSM-5 (Fig. 5f)[28,40,49].

**Control catalytic experiments and related solid-state NMR spectroscopy.** To unravel the influence of RWGS-derived CO during the reaction, we performed additional control experiments with (i) $^{12}$C$_2$H$_4$ and (ii) $^{13}$CO + $^{12}$C$_2$H$_4$ under our usual "hydrogen-rich" reaction conditions (Fig. 6) on the standalone zeolites only: MOR, ZSM-22, FER, and ZSM-5[18,19]. In both control experiments (Fig. 6a, b), ZSM-5 fully converted ethylene, whereas MOR/ZSM-22 delivered the lowest conversion. Herein, the purpose of adding ethylene in the reactant feed is to initiate the formation of HCP-based reaction centers within the zeolite pores in the absence of a metallic catalyst[27]. In the absence of $^{13}$CO, oligomerization to C$_{5+}$ hydrocarbons and paraffins (green and blue bar in Fig. 6a) is expectedly predominant on all zeolites, while MOR selectively promoted the formation of paraffins with a negligible amount of olefins (Supplementary Fig. 14). Contrary, in the presence of both $^{13}$CO and ethylene, MOR showed the highest $^{13}$CO conversion (~4%), while both ZSM-5 and ZSM-22 gave the lowest CO conversion (~1%). The hydrocarbon distribution is consistent with the original reaction results (see Figs. 2, 3, and 6b), which implies that olefins indeed could be derived from CO.

Herein, MOR enhanced the formation of light olefins (yellow bars) at the expense of consuming CO and ZSM-22 improved the formation of heavy olefins (green bars). Similarly, FER and ZSM-5 promoted the formation of paraffins (blue bars) and aromatics (purple bars), respectively. Therefore, it is safe to conclude that zeolites alone can consume CO to control/alter the product selectivity (cf. without any metallic catalyst), through HCP-based mechanisms[27]. Hence, CO could be considered as an initiator in the current study[16,47]. Next, 1D ssNMR spectroscopy also revealed similar characteristics (Supplementary Fig. 15), where,

except FER, other DE experiments are significantly broad (Fig. 6c). In this control experiment, the $^{13}$C-enrichment has selectively been employed on $^{13}$CO only (not ethylene), and hence, the observed $^{13}$C signals in 1D DE ssNMR undoubtedly confirms the direct CO incorporation on zeolites. In FER, $^{13}$CO converted selectively to mobile species, which allowed us to perform 2D correlations experiments (Supplementary Figs. 16 and 17). In Fig. 6d, a carbonyl signal at 177.5 ppm has two clear cross peaks with $^{13}$C signals at 50.4 and 19.7 ppm, which have corresponding $^1$H resonances at 3.36 (–OCH$_3$) and 2.15 ppm (–CH$_3$), respectively. This spectral pattern is typical for methyl acetate, which again unequivocally supports ketene-based reactive intermediate during the reaction[16,31,44–47,50].

**Overall reaction process.** To "connect the dots," based on the aforementioned results, a Fe$_2$O$_3$@KO$_2$/zeolite-mediated reaction pathway is proposed in Figs. 7 and 8.

Initially, the iron phase of Fe$_2$O$_3$@KO$_2$ catalyzes RWGS reaction to produce CO from the reactant feed (CO$_2$ + H$_2$) (Fig. 7a)[14,17–19,21]. Herein, the role of KO$_2$ is to enhance the adsorption and activation of CO$_2$ (Fig. 7b), as we have previously evaluated[14]. Upon exposure to our reactant feed, KO$_2$ was transformed to (well-characterized) potassium carbonate phase[14], facilitating the FTS process via a tandem mechanism. In addition, Raman (micro)spectroscopy and air-protected capillary single-crystal X-ray diffraction studies were performed on both fresh and spent Fe$_2$O$_3$@KO$_2$ catalysts (Supplementary Figs. 18–20; also see Supplementary Discussion), which revealed that the fresh metallic catalyst was constituted by γ-Fe$_2$O$_3$ (maghemite) and potassium carbonate phases, while the spent catalyst was a complex mixture of χ-Fe$_5$C$_2$ (the active FTS phase of the catalyst), γ-Fe$_2$O$_3$, and numerous K-based inorganic carbonyl salts, including potassium carbonate (K$_2$CO$_3$), potassium bicarbonate (KHCO$_3$), and potassium formate (KOOCH). Furthermore,

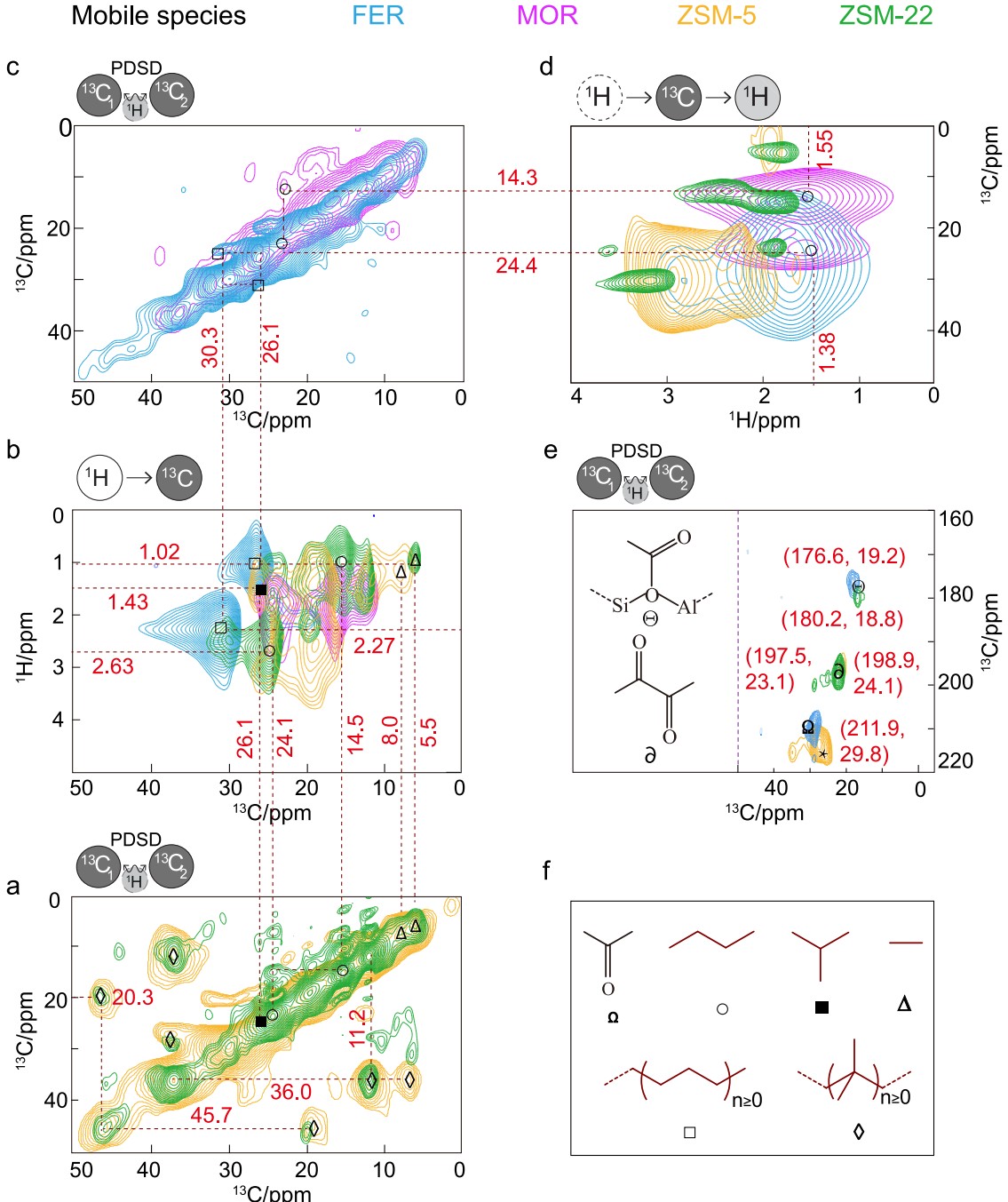

**Fig. 4 Identification of postreacted zeolite-trapped mobile molecules by 2D MAS solid-state NMR spectroscopy.** $^{13}C$-$^{13}C$ correlations in the aliphatic regions on postreacted zeolites **a** ZSM-5 and ZSM-22 as well as **c** MOR and FER. **b** $^{1}H$-$^{13}C$ INEPT HETCOR and **d** $^{13}C$-$^{1}H$ HSQC spectra of the aliphatic region from all four postreacted zeolites. **e** $^{13}C$-$^{13}C$ correlations in the carbonyl regions on postreacted zeolites FER, ZSM-5, and ZSM-22, highlighting the presence of ester and ketones (*: spinning sidebands on ZSM-5). See Supplementary Fig. 12 for the ssNMR spectra of these chemisorbed carbonylated species to verify the respective assignments. **f** Identified molecular scaffolds. To probe mobile $^{13}C$-$^{1}H$ correlations, "through-bond" scalar magnetization transfer was used to polarize the carbons (in **b** and **d**), whereas in the $^{13}C$-$^{13}C$ correlation spectra, the carbons were polarized through direct excitation, and $^{13}C$-$^{13}C$ mixing was achieved through proton-driven spin diffusion (PDSD) (in **a**, **c**, and **e**). Spectra of trapped products obtained on respective postreacted zeolites after the hydrogenation of fully isotope-enriched $^{13}CO_2$ in the reactant feed ($^{13}CO_2$ at 30 bar, 375 °C, $H_2/^{13}CO_2 = 3$, and 10,000 mL·g$^{-1}$·h$^{-1}$ at a time on stream of 48 h). The respective full-range spectra, as well as experimental details, are included in Supplementary Information (MAS magic angle spinning, HETCOR HETeronuclear CORrelation spectroscopy, INEPT insensitive nuclei enhanced by polarization transfer, HSQC heteronuclear single quantum coherence spectroscopy).

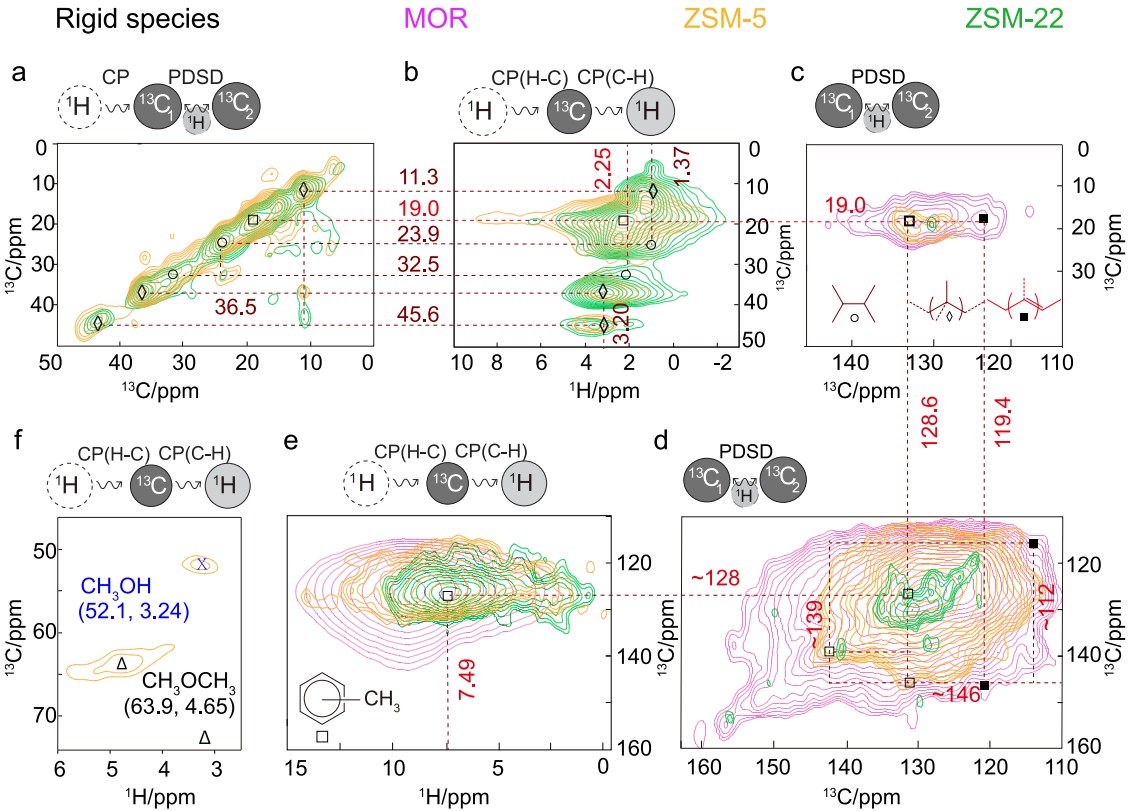

**Fig. 5 Identification of postreacted zeolite-trapped rigid molecules by 2D MAS solid-state NMR spectra. a** $^{13}C$-$^{13}C$ and **b** $^{13}C$-$^{1}H$ correlations in the aliphatic regions on postreacted zeolites ZSM-5 and ZSM-22. The identification of methylated aromatic/olefinic species via **c** and **d** $^{13}C$-$^{13}C$ and **e** $^{13}C$-$^{1}H$ correlations on zeolites ZSM-5, ZSM-22, and MOR. **f** $^{13}C$-$^{1}H$ correlations highlighting methoxy region only, confirming the presence of methanol/DME. Identified molecular scaffolds were represented in symbols (MAS magic angle spinning). Herein, for the $^{13}C$-$^{1}H$ correlation spectra (in **b**, **e**, and **f**) "through-space" dipolar cross polarization was used to polarize the carbons, whereas in the $^{13}C$-$^{13}C$ correlation spectra, the carbons were polarized either through cross polarization (in **a**) or direct excitation (in **c** and **d**) and $^{13}C$-$^{13}C$ mixing was achieved through proton-driven spin diffusion. Spectra of trapped products obtained on respective postreacted zeolites after the hydrogenation of fully isotope-enriched $^{13}CO_2$ in the reactant feed ($^{13}CO_2$ at 30 bar, 375 °C, $H_2$/$^{13}CO_2 = 3$, and 10,000 mL·g$^{-1}$·h$^{-1}$ at a time on stream of 48 h). Respective full-range spectra and experimental details are included in Supplementary Information.

based on the aforementioned analysis, the interconversion of $K_2CO_3$ and $KHCO_3$/$KOOCH$ under our reaction conditions was demonstrated to lead to methanol/DME under hydrogenation conditions (Figs. 5f and 7b)[51]. Therefore, methanol could be produced during the hydrogenation of inorganic carbonates over $Fe_2O_3@KO_2$[10,21,51] and/or as a part of CO-insertion mechanism (Fig. 7c, d) (vide infra). Moreover, alcohols have long been hypothesized to be chain initiators during the Fe mediated FTS process, based on $^{14}C$-radiotracer experiments[52,53]. On this note, the absence of methanol in controlled experiments further confirms the hypothesis that methanol indeed was formed on $Fe_2O_3@KO_2$ catalyst under FTS condition (Fig. 6 and Supplementary Fig. 17). During FTS-based chain initiation over the iron catalyst, the activation of CO could happen either via dissociative/carbide pathway (i.e., Biloen–Sachtler mechanism)[53,54] and nondissociative/CO-insertion pathway (i.e., Pichler–Schulz mechanism) (Fig. 7c)[53,55]. Among them, the carbide pathway is the most accepted mechanism to date, while in this work, we have provided significant evidence supporting the CO-insertion pathway as well[53]. Similarly, both carbide and CO-insertion mechanisms were operational during the chain growth to produce FTS products (Fig. 7d). Since the dual-bed system is implemented, the entire gas stream (CO, methanol, and typical FTS-derived hydrocarbons) has been passed to the zeolite phase from the metallic phase. Interestingly, it should be emphasized again that organic-carbonylated species were identified exclusively on the

zeolite phase (i.e., lower-catalyst bed; see Fig. 4, Supplementary Discussion, and Supplementary Figs. 12 and 13), whereas K-based inorganic carbonylated salts were detected only on the metallic phase (i.e., upper-catalyst bed; Supplementary Figs. 12, 13, and 18–20). Upon developing a carbonylative environment, a part of FTS hydrocarbons converted to zeolite acetate/methyl acetate, diacetyl, and acetone (with ketene being a potential reactive intermediate[16,28,31,44–47,56]) (Supplementary Fig. 24 and Fig. 8 below). Next, the formation of shorter olefins was initiated by the FTS process, which is further improved by MOR via additional consumption of RWGS-derived CO.

Next, the zeolite phase governs the ultimate product selectivity during the reaction, as was emphasized throughout the manuscript. In the HCP route, shorter olefins oligomerized in ZSM-22 to produce longer (olefinic) hydrocarbons, while hydrogen transfer reactions led to paraffins and aromatics in FER and ZSM-5, respectively. In addition, in ZSM-5, relatively wider straight channels endorse the cyclization of oligomerized compounds and subsequent formation of aromatics, which is nonattainable on FER or ZSM-22[27,30,57]. Due to the molecular size of zeolite pores and the numerous topologies available, the structure of the hydrocarbons can be "molded" to a specific type. Olsbye et al. performed a comprehensive study at 400 °C and 80% methanol conversion to demonstrate that 1D large pore zeolites, such as ZSM-22 (TON with 10-ring elliptical channel) and ZSM-23 (MTT with 10-ring teardrop channel), could deliver $C_{5+}$

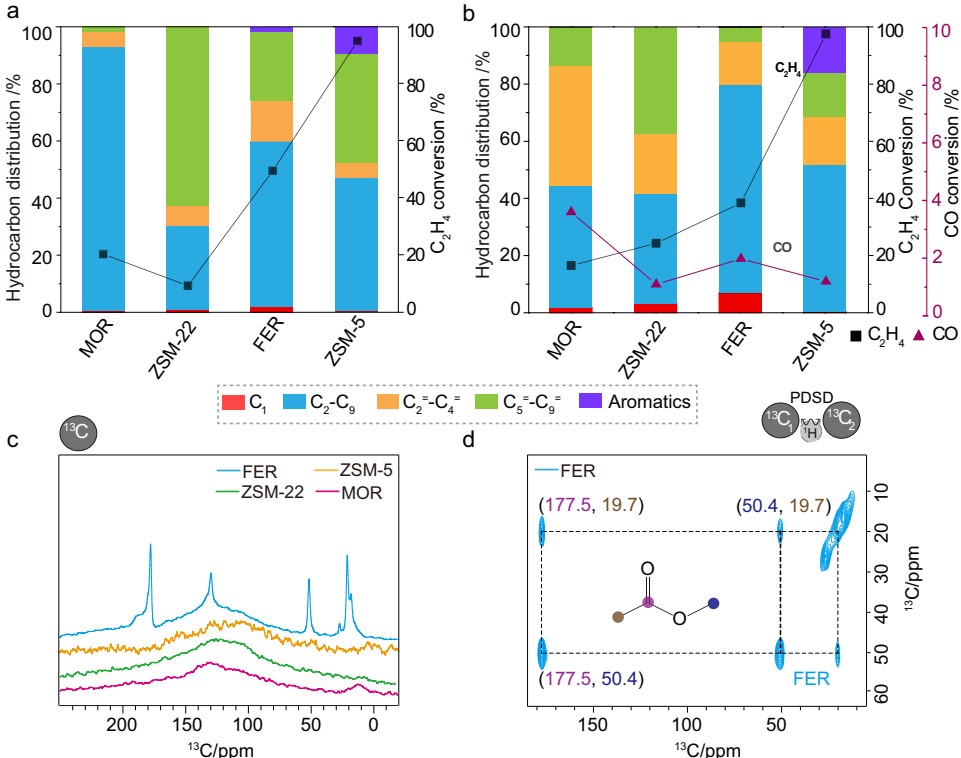

**Fig. 6 A summary of control experiments.** The catalytic performance of the standalone zeolite-mediated conversion of **a** ($^{12}$C)ethylene and **b** fully enriched $^{13}$CO and ($^{12}$C)ethylene at 30 bar, 375 °C, and 10,000 mL·g$^{-1}$·h$^{-1}$ under hydrogen-rich environment at a time on stream of 2 h (i.e., without any Fe$_2$O$_3$@KO$_2$ phase) (also see Supplementary Fig. 14). **c** 1D $^{13}$C direct excitation MAS solid-state NMR spectra on all four zeolites, collected after the control experiments involving both $^{13}$CO and $^{12}$C$_2$H$_4$ in the reactant feed. Due to the selective isotope enrichment of $^{13}$CO in the reactant feed, DE experiments can only be correlated to the $^{13}$CO consumption during catalysis. The broad nature of DE experiments (except FER) implying that $^{13}$CO predominantly converted to the rigid species only (also see Supplementary Figs. 15 and 16 for details, MAS magic angle spinning). **d** The spectral identification of methyl acetate was observed on the postreacted FER zeolite via $^{13}$C-$^{13}$C correlation spectra, where the carbons were polarized through direct excitation and $^{13}$C-$^{13}$C mixing was achieved through proton-driven spin diffusion (PDSD) (also see Supplementary Fig. 17). The respective full-range spectra, as well as experimental details, are included in Supplementary Information.

aliphatics products without any aromatics[58–60]. To produce aromatics with 10-member-ring zeolites, ZSM-5 (MFI 3D 10-ring channel with cross sections) is more appropriate[30], as also demonstrated in the current study. Furthermore, an additional factor necessary to focus on when explaining product shape selectivity, apart from the overall zeolite topology, is the structure of the individual pores forming the zeolite. Recent studies demonstrated that the sinusoidal channel of H-ZSM-5 favors the olefin/paraffin cycle, whereas the straight channel facilitates the production of aromatics[27,30,57], which is also evidenced by the higher selectivity of ZSM-5 toward aromatics and paraffins in our experiments. Moreover, theoretical calculations conducted on a wide range of hydrocarbons and zeolites, including those previously mentioned, highlighted that the diffusion rate of hydrocarbons through zeolite pores is dependent on the channel's height, width, and shape[61,62], i.e., the molecular fit. Furthermore, validating that product outflow is dependent on the structural inhibition imposed by the molecular size and shape of the zeolite pores and the crucial role of topology as a product shape selectivity descriptor. Based on our results, both larger and smaller pores are not able to form the required transition states under our reaction conditions. Specifically, 8 MR (CHA, DDR) and 12 MR (MOR, BEA, FAU) are not providing the right fit under the given conditions in our system, whereas 10 MR zeolites (MFI, TON, FER) are the only ones inducing the right confinement effects leading to oligomerization and further

production of hydrocarbons, as previously investigated in various zeolite catalysis (Fig. 7e)[63,64].

Although the zeolite lattice ensures the necessary space to form a specific hydrocarbon, several studies showed that zeolites with identical topologies can perform differently due to an uneven acid site strength[58,65,66]. In addition to acid strength, acid site density was proven to influence the shape and reactivity of the "hydrocarbons pool" and thus, the overall product distribution[58,67,68]. Recently, Chowdhury et al. demonstrated the formation of (MTH-like) shorter olefins and (olefinic and aromatic) HCP species exclusively from acetyl group over H-SAPO-34 catalyst, without any involvement of methanol[31]. Similarly, acetone/ketones are already reported to initiate the formation of initials C–C bonds and HCP species over zeolite[69,70]. Therefore, these HCP-based organic species could independently initiate the formation of hydrocarbons (via promoting C–C bond couplings)[27,28,31–33] and together with the acidity within the zeolite are considered analogous to "hybrid inorganic–organic supramolecular reaction centers"[34–36], similarly encountered in MTH chemistry[27,31–33], and hence, act as "descriptors" to regulate the final product selectivity (Fig. 7f).

**Theoretical analysis of zeolite phase chemistry.** Mechanistically, the most unique feature is the spectroscopic identification of multiple organic-carbonylated species (Fig. 4e), which could

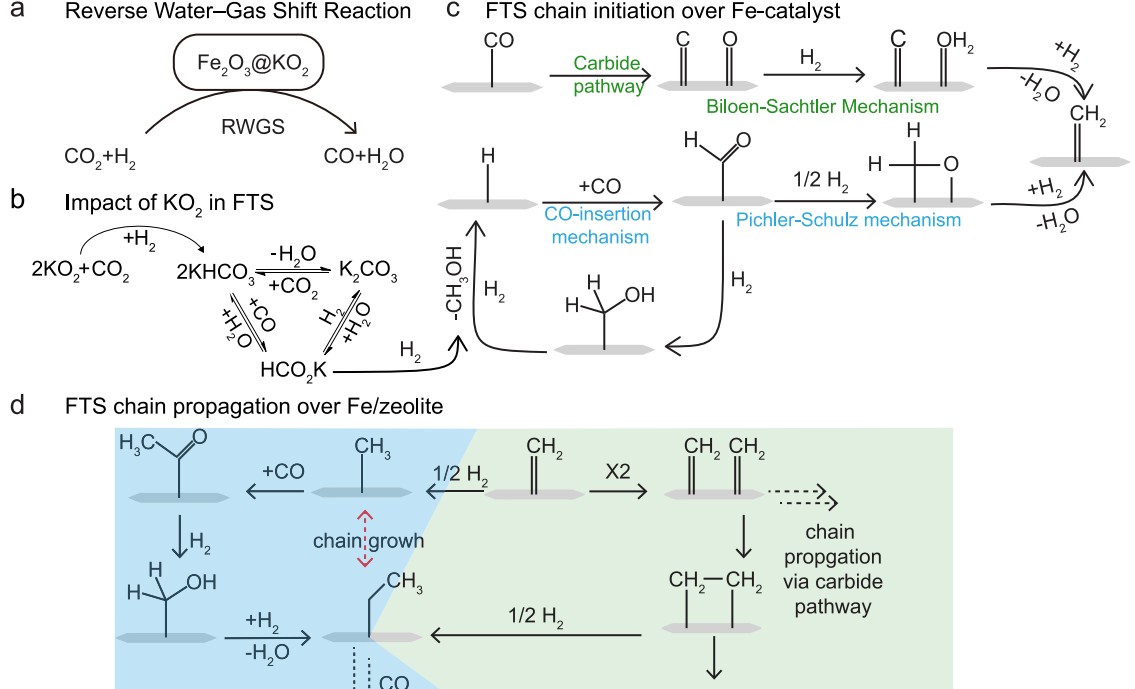

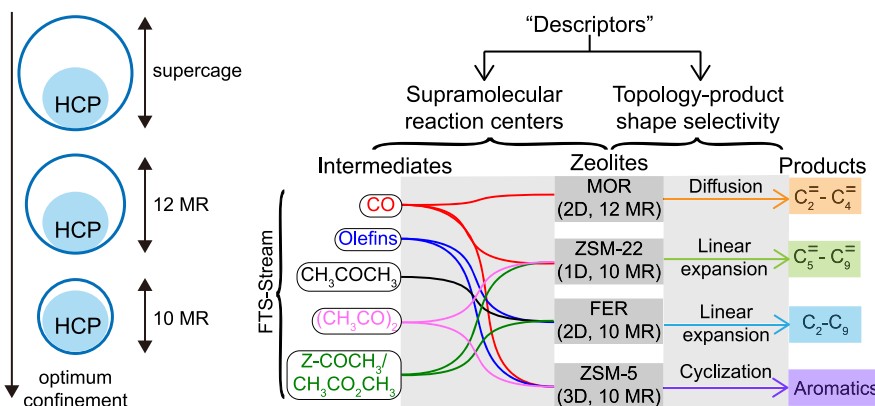

**Fig. 7 The proposed mechanistic pathway for the Fe₂O₃@KO₂/zeolite catalyzed hydrogenation of CO₂, highlighting the efficacy of our multifunctional system.** The reaction sequence on **a** iron and **b** potassium phase of Fe₂O₃@KO₂ material implying the significance of tandem activation of CO₂ in this work. FTS-based **c** chain initiation and **d** chain propagation steps through both carbide and CO-insertion pathways; also known as Biloen–Sachtler mechanism (greenish background) and Pichler–Schulz mechanism (bluish background), respectively. **e** Simplified illustration highlighting the concepts of "molecular fit" and "containment effect" in zeolite catalysis. **f** A simplified relationship sketch between hydrocarbons/carbonylates (including CO) and final product selectivity for all zeolites, highlighting the significance of both supramolecular reaction centers and topology on controlling the product selectivity. (RWGS reverse water gas shift reaction, FTS Fischer–Tropsch Synthesis, HCP hydrocarbon pool).

directly be linked to ketenes, an intermediate recently attracting attention for its influential role in zeolite catalysis[16,28,31,44–47]. However, the direct spectral identification of ketene requires highly sophisticated analytical techniques (as reported elsewhere[16,44]), and impossible to identify on zeolites. Ketenes could easily be physisorbed on BAS of zeolite to produce surface-acetate species (i.e., a protonated ketene, $CH_2CO + H^+ \rightarrow CH_3CO^+$)[31], with a large energy gain of >50 kJ/mol and a minimal energy barrier of ≤17 kJ/mol[46]. Therefore, such a quick equilibrium toward acetate, along with ketene's low steady-state concentrations at high reaction temperatures, forbids its direct spectroscopic identification upon zeolite. Not only acetate, acetone and diacetyl could also be reversibly accessible from ketenes at high temperature[48]. Herein, diacetyl

is the direct C–C bond-forming product from acetate/ketene (cf. conceptually similar to "CO-dimerization pathway")[5].

The above hypotheses are further confirmed by means of computational calculations (Table 1, Fig. 8, and Supplementary Figs. 21–28; also see Supplementary Discussion for further in-depth analysis). Depending on the orientation of ketene to the zeolite acid site, small differences in the adsorption energies, relative to the more stable adsorption configuration via the C moiety, were calculated (−3 to 12 kJ/mol). This ensures a broader scope of reaction routes for ketene, via the C or O atom (see Supplementary Figs. 23 and 24). We observed that ketene bonding is slightly more exothermic via the carbon center adsorption; indicating a preference for the formation of acetyl

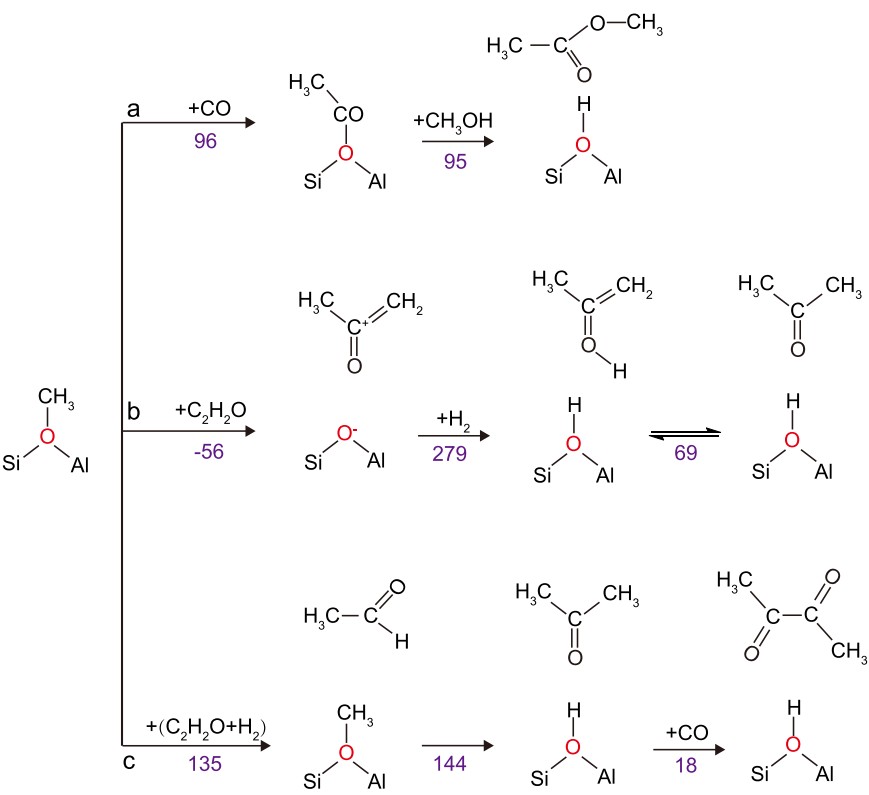

**Fig. 8 Illustration of proposed formation routes of carbonylated intermediates. a** The formation of surface acetate and methyl acetate from surface-methoxy species and two acetone production routes: **b** by direct conversion of ketene via keto-enol tautomerization as well as **c** initial hydrogenation of ketene to acetaldehyde and diacetyl from acetone, conducted on FER models, with reaction energies (in violet) presented as -($E_r$) in kJ/mol.

**Table 1 Summary of energetic observables.**

|  | ZSM-22 [T1] | ZSM-5 [T12] | ZSM-5 [T6] | MOR [T1] | FER [T3] | FER [T1] |
|---|---|---|---|---|---|---|
| $-E_{ads}$ |  |  |  |  |  |  |
| CO | 49 | 49 | 49 | 45 | 49 | 37 |
| Ketene (O) | 167 | 160 | 167 | 155 | 163 | 159 |
| Ketene (C) | 175 | 171 | 164 | 164 | 175 | 165 |
| Methyl acetate | 144 | 144 | 138 | 127 | 138 | 135 |
| Diacetyl | 134 | 141 | 111 | 120 | 132 | 124 |
| Acetone | 147 | 141 | 110 | 127 | 126 | 134 |
| $-E_{bond}$ |  |  |  |  |  |  |
| $Z$-COCH$_3$ | 995 | 931 | 919 | 927 | 966 | 965 |
| $-E_r$ |  |  |  |  |  |  |
| Ketene + $Z$-H$^+$ → $Z$-COCH$_3$ | 31 | 46 | 44 | 37 | 31 | 47 |

Adsorption energies ($E_{ads}$) of molecular species to the BAS of zeolites, of the heterolytic dissociation energy of the zeolite-acetyl bond ($E_{bond}$), and reaction energy ($E_r$) to form acetyl –$Z$-COCH$_3$ from proton addition to ketene, presented in kJ/mol, with ketene models as ketene bonded to the active site via oxygen (O) or carbon (C). For simplicity, all energetic observables are presented as: -($E_{ads}$), -($E_{bond}$), -($E_r$).

species through the reaction between ketene and zeolitic Brønsted protons[16,31,44,45]. The presence of acetate species in both ZSM-22 and FER could be rationalized in terms of its higher bonding energy than others. The relatively higher instability of any carbonylates in H-MOR[T1] implies that the reaction equilibrium would be slightly more shifted toward CO, consistent with both spectroscopic and control experiments (Figs. 4e and 6b). In addition to the ketene, CO insertion has emerged as an alternative route based on energetics involved. As previously reported and will be further discussed (vide infra), CO insertion to surface methoxy (reaction energy, $E_r = 93$ kJ/mol)[50] is more exothermic than ketene conversion ($E_r \sim 40$ kJ/mol) on ZSM-5[T12], with more detailed explanations of these observations presented in the computational subsection of Supplementary Discussion.

theoretical investigations also indicated the possibility of a competitive adsorption equilibrium between acetone, methyl acetate, and diacetyl, also further detailed in in the computational subsection of Supplementary Discussion and Supplementary Fig. 25. Owing to their similar stability, the selective experimental observation in our ssNMR investigation may indicate preferential diffusion or confinement effects of certain carbonylates, emphasizing the direct relevance of host–guest chemistry during catalysis.

To further clarify the chemistry inside the zeolite pores, several reaction routes describing the formation of the carbonylated intermediates were analyzed from a thermodynamic perspective. Since FER stabilized a broad range of carbonylated species while having a small number of acid sites to dominate the adsorption

process, the main focus on determining the favorable energetics involved in producing the main intermediates was based on the FER[T1] models. As presented in Fig. 8a, both the formation of surface acetate from surface-methoxy species ($E_r$ = 96 kJ/mol) and methyl acetate from surface acetate ($E_r$ = 95 kJ/mol) are thermodynamically favored, as previously reported[47,50]. In the case of acetone formation, two reaction routes starting from ketene were proposed. The first starts with the endothermic formation of methylated ketene ($E_r$ = −56 kJ/mol; Fig. 8b), followed by exothermic hydrogenation ($E_r$ = 279 kJ/mol; Fig. 8b) and isomerization ($E_r$ = 69 kJ/mol; Fig. 8b). The second route starts with the hydrogenation of ketene to acetaldehyde ($E_r$ = 135 kJ/mol; Fig. 8c) and further methylation ($E_r$ = 144 kJ/mol; Fig. 8c) to acetone, both steps occurring with a considerably high energy release. Additional carbonylation of acetone to diacetyl was also shown to be a viable reaction route ($E_r$ = 18 kJ/mol; Fig. 8c). Since all intermediates are formed through exothermic reaction routes, the pathway of Fig. 8c is expected to be highly feasible and further contribute to the initiation of $C_{2-4}$ hydrocarbons production[34,71,72]. Nevertheless, considering the high temperatures at which this reaction is performed, the pathway of Fig. 8b represents another mechanistic option, contributing to the complexity of the intermediates in the reaction pool.[31,69,70].

## Discussion

The combination of metal catalysis and different zeolite frameworks opens the door to a wide range of petrochemical products from the direct hydrogenation of $CO_2$. In this work, we have rationalized the influence of the zeolite framework by building a complete catalytic database involving eight different zeolites in combination with a $Fe_2O_3$@$KO_2$ catalyst. As a result, four different selectivity patterns can be derived specific to zeolite topology features: light olefins (MOR, SAPO-34, ZSM-58, BETA, Y) and to a greater selectivity improvement toward paraffins (FER and ZMS-5), long olefinic hydrocarbons (ZSM-22), and aromatics (ZMS-5). The main influence on product selectivity is observed to be determined by  10 MR zeolites (TON, FER, and MFI), as opposed to smaller pores 8 MR (CHA, DDR) and larger pores 12 MR (MOR, BEA, FAU) zeolites, as 10 MR zeolites are known to induce the optimum confinement effects leading to oligomerization and conversion of the hydrocarbons pool. An in-depth spectroscopic and computational analysis along with experiments with fully enriched $^{13}CO_2$ and $^{13}CO$ collectively revealed that such preferential origin of selectivity could also be attributed to the formation of carbonylated species (including RWGS-derived CO or further CO-derived ketene/ketone/ester) that together with the "inorganic" zeolite form HCP-based catalytic centers that further activate the outflow of compounds from the iron phase. Interestingly, numerous organic-carbonylated intermediates have been spectroscopically identified, possibly for the first time during zeolite-mediated thermal $CO_2$ catalysis. The hybrid nature of the active zeolite catalyst, i.e., the inorganic–organic supramolecular reactive centers along with zeolite topologies, are shown to act as product selectivity descriptors, in the current study, and are expected to be used to design the next generation catalyst applied for the adsorption and conversion of $CO_2$ into different industry relevant materials.

## Methods

The chemicals, zeolites, and gases were purchased from Aldrich, Alfa Aesar, Zeolyst, ACS materials, and CK Isotopes Limited. Catalytic tests were executed in a 16 channel Flowrence® from Avantium, connected to Agilent 7890B gas chromatographic system for the analysis. Nitrogen adsorption and desorption isotherms were recorded on a Micromeritics Asap 2420 at 77 K. The temperature-programmed ammonia desorption ($NH_3$-TPD) experiments were carried out in an AMI-200ip Catalyst Characterization System (Altamira) equipped with TCD. All

$^1H$ and $^{13}C$ related (both 1D and 2D) MAS ssNMR spectroscopic experiments were performed on Bruker AVANCE III spectrometers operating at 400 MHz frequency for $^1H$ using a conventional double resonance 3.2 mm CPMAS HX probe (CP). $^1H$ and $^{13}C$ NMR chemical shifts are reported with respect to the external reference adamantane. $^{27}Al$ MAS ssNMR experiments were carried out on a 900 MHz Bruker AVANCE IV 21.1 T spectrometers quipped with 3.2 mm CPMAS probes, where chemical shifts were externally referenced to $Al(NO_3)_3$. All NMR measurements were performed at room temperature (298 K) and MAS frequency of 16 or 20 kHz (unless specified otherwise in the figure captions). All NMR spectra were processed and analyzed using Bruker TopSpin 4.0. Raman spectra were recorded using a confocal Raman microscope WITec Apyron equipped with 532 and 633 nm laser lines. Powder X-ray diffraction measurements for air sensitive catalyst were conducted using a Bruker D8 Venture single-crystal diffractometer equipped with a PHOTON II area detector and an IμS microfocus source (set to 50 kV, 1 mA) providing an Mo Kα radiation (λK1 = 0.70930 Å, λK2 = 0.71359 Å). Computational calculations were conducted using Vienna ab initio Simulation Package[73,74], Perdew–Burke–Ernzerhof functional with Grimme's dispersion correction (PBE-D3)[71], and a plane-wave basis set of the projector-augmented-wave method[72]. The adsorption energy ($E_{ads}$) of an adsorbate (CO, ketene, methyl-acetate, acetone and diacetyl), bonding energy ($E_{bond}$) and reaction energy ($E_r$) are calculated as follows: $E_{ads}$ = $E_{[ZeOH+Sorbate]}$ − $E_{[ZeOH]}$ − $E_{[Sorbate]}$ (1) where, $E_{[ZeOH]}$, $E_{[Sorbate]}$ and $E_{[ZeOH+Sorbate]}$ are the total energy of the zeolite sorbent, the neutral gas-phase sorbate and the combined guest-host system, respectively, each in their optimised geometry. The bonding energy ($E_{bond}$) for a zeolite bonded moiety (surface-acetate) was calculated as: $E_{bond}$ = $E_{[ZeO+moiety]}$ − $E_{[ZeO-]}$ − $E_{[moiety+]}$ (2) where, $E_{[ZeO-]}$, $E_{[moiety+]}$ and $E_{[ZeO+moiety]}$ are the total energy of the deprotonated (i.e., anionic) zeolite, the cationic gas-phase moiety and the combined guest-host system, respectively, each in their optimised geometry; $E_r$ = $E_{[PR]}$ − $E_{[R]}$(3) where, $E_{[PR]}$ and $E_{[R]}$ are the absolute energies of the product and reactant states, each in their optimized geometry. For simplicity, all energetic observables are discussed and presented as −($E_{ads}$), −($E_{bond}$), −($E_r$). Finally, we refer to Supplementary Methods in Supplementary Information for a detailed description.

## Data availability

All data that support the findings of this study are available within the paper and its Supplementary Information or from the corresponding authors upon reasonable request.

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

## Acknowledgements

This project has received financial supports from the King Abdullah University of Science and Technology (Saudi Arabia), the start-up research grant from the Institute for Advanced Studies (IAS), Wuhan University (China), and the National Natural Science Foundation of China (NSFC) (Grant No. 22050410276 to A.D.C.). The authors also acknowledge Mr. Ye Yiru (IAS, Wuhan, China) and Dr. Serhii Vasylevskyi (KAUST, Saudi Arabia) for their support during figures' preparation and Raman measurements, respectively. A.D.C. also conveys sincere thanks to Dr. Alessandra Lucini Paioni for her advice in this work.

## Author contributions

J.G. and A.D.C. conceived the research ideas and supervised the overall project. A.R. and A.D.C. designed the experiments. A.R. performed the catalytic experiments. A.D.C. and E.A.-D. performed NMR experiments, and X.G. and A.D.C. did data analysis. M.C. did all other characterization. S.-A.N. and L.C. performed and supervised the computational analysis. L.G. worked with reactor and catalytic testing. A.D.C., X.G., A.R., S.-A.N., and J.G. cowrote the original draft. All authors discussed the results as well as contributed and commented on the different versions of the manuscript.

## Competing interests

The authors declare no competing interests.
