## [Peer Review File · Nature Communications]

REVIEWER COMMENTS

Reviewer #1 (Remarks to the Author):

Catalytic conversion of CO₂ into high-value chemicals is a promising route for the utilization of CO₂ and global warming mitigation. Cascade process combining the advantage of metal catalysts and zeolites has attracting increasing interest in C₁-feedstocks conversion. However, the catalytic reaction is still poorly understood. In this work, the authors investigated direct hydrogenation of CO₂ over Fe₂O₃@KO₂ catalyst in combination with eight different zeolites. Advanced solid-state NMR spectroscopy coupled with computational analysis were used to study the reaction mechanism. The 2D NMR experiments provide a wealth of information on the species trapped in zeolites. The different selectivity to olefins and hydrocarbons was attributed to the favored formation of carbonylated intermediates and hydrocarbon species in different zeolites. The work is interesting and the paper is well organized. The paper is publishable after the following issues have been addressed.

1. How about the role of KO₂ in the catalytic conversion of CO₂? The authors should give some discussion on this point.
2. As we known, Fe often shows paramagnetic character, which can raise the difficulty for the NMR detection of the species nearby. The analysis of the possible intermediates around the Fe-phase in the combined catalyst would be helpful to obtain comprehensive knowledge of the CO₂ conversion.
3. In Figure S3, why were the carbonyl species only detectable in the DE (direct excitation) experiments, but not in CP? In addition, the CP signal intensities of aromatics were weaker than in DE experiments. why?
4. Identification of exact species formed in a complicated reaction is not a trivial work. The authors attributed the ¹³C signals at 197.5/23.05ppm to diacetyl on ZSM-5. Alternatively, these signals can also come from other species. Complementary experiments are expected to consolidate the assignments.
5. In figure 7, the authors proposed that carbonlyation of (CH₂)_n could generate ketene. How does this route operate? In addition, since the C-C hydrocarbons were formed, was it necessary to continue to proceed secondary reaction to form the so-called intermediate? It seems that carbonylation of methoxy species (generated by methanol) is a more reasonable route for the formation of initial C-C bond species according to the previous work (Angew. Chem. Int. Ed. 2006, 45, 1617-1620; J. Phys. Chem. C 2013, 117, 5840-5847; Angew. Chem. Int. Ed. 2016, 55, 5723-5726). In this mechanism, methyl acetate and acetyl were usually formed and the latter could be readily converted to ketone which was unstable (also not observed in this work) and could be further transformed to other species. Although methanol was solely observed by the authors on ZSM-5, the absence of methanol on other zeolites does not mean this route is not involved in the generation of ketone and related species.
6. The authors proposed the C-C coupling of methoxyl with ketene led to the formation of acetone. I am confused that how this route operates? Is hydrogen needed in this route? In addition, what is the meaning of "2*"?
7. In this work, the authors introduced the "organic-inorganic" hydrid nature of the working zeolite, namely "supramolecular reactive center", which had been proposed and investigated in the MTH reaction by previous work (J. Am. Chem. Soc. 2000, 122, 10726-10727; Angew. Chem. Int. Ed. 2016, 55, 2507-2511).

Reviewer #2 (Remarks to the Author):

In this work, "Selectivity descriptors for the direct hydrogenation of CO₂ to hydrocarbons during zeolite-mediated bifunctional catalysis", the authors provided a systematic study and carried out detailed characterization to understand the descriptors for the direct hydrogenation of CO₂ to hydrocarbons during zeolite-mediated bifunctional catalysis. There have been many papers published

regarding metal oxide/zeolite composite catalysts to control the product selectivity including light olefins, branched hydrocarbons, and aromatics. In this study, the authors used solid-state NMR spectroscopy and computational method to demonstrate the hybrid nature of the zeolite catalyst to produce different types of reaction intermediates for the ultimate product selectivity. It is an appreciable work for the catalysis research. However, the work does not have much more novelty to publish in Nature communications and the reasons are below.

1. The manuscript describes the influence of the zeolite framework, involving eight different zeolites, in combination with a Fe₂O₃@KO₂ catalyst and depending on the zeolite, various products such as light olefins (MOR, SAPO-34, ZSM-5, BETA, Y), paraffins (FER), long (olefinic) hydrocarbons (ZSM-22), and aromatics on (ZMS-5) were produced depending on the zeolite framework; it is well-known phenomena in the CO₂ hydrogenation. The authors are trying to rationalize the formation of intermediates/descriptors of the formation of hydrocarbons on Fe₂O₃@KO₂/zeolite. However, the authors are already published the data in their previous publication; ACS Catal. 2019, 9, 7, 6320–6334; using ZSM-5 and MOR with Fe₂O₃@KO₂. Solid-state nuclear magnetic resonance characterization of the zeolites revealed that the reaction mechanism is driven by the incorporation of CO in the network in the form of surface formate. Whereas, in this manuscript, the authors presented CO formed from the RWGS further converted to ketene/ketone/esters on the same Fe₂O₃@KO₂/zeolite. It is contradictory results.
2. Identification of trapped molecules using 2D MAS solid-state NMR in the zeolite pores were previously reported in a number of literatures (e.g., ACS Catal. 2018, 8, 7356–7361). In addition, the formation of surface acetate, formate and methylated polyene/benzene in H-SAPO-34 was reported previously (Angew. Chem. Int. Ed. 2016, 55,15840-15845).
3. The authors describe CO is a co-catalyst in the reaction. We think that CO is not a co catalyst and it is the initiator for the hydrocarbon production. In page no 17, the authors reported that 13CO predominantly was converted to rigid species. Then how can assume CO is a co-catalyst?
4. The reaction mechanism in page no 20 (Fig.7a); the role of Fe₂O₃@KO₂ is only RWGS using MOR zeolite? It is difficult to believe! Because during the reaction Fe₂O₃ converted to Fe₅C₂ and C-C coupling reactions takes place on it?
5. It is necessary to present ssNMR spectrum of the spent standalone Fe₂O₃@KO₂ catalyst and compare with ssNMR spectra of the Fe₂O₃@KO₂/zeolite. Based on recent study on in situ FT-IR of Fe-based catalysts, some of reaction intermediates are present in the Fe-based catalysts. Thus, the reaction intermediates that were discussed in the ssNMR spectra of the Fe₂O₃@KO₂/zeolite may not solely from the zeolite.
6. The authors presented the reaction mechanism for the formation of methanol in Fig. 7b. However, there is no reaction data or ssNMR spectrum to prove the presence of methanol in either the manuscript or in the supplementary data. How do you confirm the addition of methanol in the ketene on the zeolite?
7. Most of the ssNMR spectra are the trapped molecules in the spent zeolite. Since these compounds are physically adsorbed, and not further converted during the reaction, how can say these are descriptors for the CO₂ hydrogenation reaction ? Really these species are descriptors or spectators ?
8. In the computation study, full reaction pathways with activation energies, comparison of reaction coordinates of possible reaction pathways should be presented.

Reviewer #3 (Remarks to the Author):

The authors synthesized the Fe-based catalysts for CO₂ hydrogenation to olefins, aromatics and paraffins. It is interesting that the reaction intermediates in different zeolites were characterized. Different zeolite and reaction intermediates were found to govern the ultimate product selectivity. However, some questions need to be addressed before its publication, the details are as follows:

1. The title of this manuscript is "selectivity descriptors for the direct hydrogenation of CO₂ to ...", however the detail "descriptors" for the selectivity of CO₂ hydrogenation was not clearly described.
2. The manuscript gave the detailed information for the reaction intermediates in different zeolites by ssNMR and the calculation. I think the reaction intermediates and product were well characterized. However, it only gave a phenomena, but not the intrinsic reason.
3. The eight different zeolite topologies were classified into four groups based on the selectivity, so the topology should be the descriptor, then why the zeolites with the same topology showed different selectivity?

Point-by-Point Response to Reviewers' Comments

Reviewers' comments are written in **blue**

Authors' responses are in **black**

Reviewer #1 (Remarks to the Author):

Catalytic conversion of CO₂ into high-value chemicals is a promising route for the utilization of CO₂ and global warming mitigation. Cascade process combining the advantage of metal catalysts and zeolites has attracting increasing interest in C1-feedstocks conversion. However, the catalytic reaction is still poorly understood. In this work, the authors investigated direct hydrogenation of CO₂ over Fe₂O₃@KO₂ catalyst in combination with eight different zeolites. Advanced solid-state NMR spectroscopy coupled with computational analysis were used to study the reaction mechanism. The 2D NMR experiments provide a wealth of information on the species trapped in zeolites. The different selectivity to olefins and hydrocarbons was attributed to the favored formation of carbonylated intermediates and hydrocarbon species in different zeolites. The work is interesting and the paper is well organized. The paper is publishable after the following issues have been addressed.

Response: We thank reviewer 1 for the encouraging comments, suggestions, and positive evaluation. Our responses with respect to the specific queries are discussed below.

1. How about the role of KO₂ in the catalytic conversion of CO₂? The authors should give some discussion on this point.

Response: Thank you for pointing this out. We apologize that we did not add sufficient information regarding the role of KO₂ before, although it was somewhat highlighted in Fig. 7b. Therefore, we have now completely re-written the mechanistic part under the newly added "Overall Reaction Process" section (Pages 16-23 in the manuscript, including Fig. 7-8). In essence, we have previously demonstrated that the active iron catalyst is typically encapsulated in K₂CO₃, facilitating the formation of olefins from CO₂ via a tandem mechanism (see ref. 14

in the revised manuscript)¹. In the revised manuscript, we have also included additional experimental supports to provide more in-depth analysis using Raman (micro)spectroscopy and air-protected capillary single-crystal X-ray diffraction studies on both fresh and spent Fe₂O₃@KO₂ catalysts (Supplementary Fig. 18-20; also see Supplementary Section S2.1 for the detailed discussion). Such K-doping is necessary to introduce an ideal balance between iron-oxide (to perform RWGS) and iron-carbide (to perform Fischer-Tropsch). As illustrated in Fig. 7b, upon exposure to 'CO₂+H₂' feed, KO₂ immediately transformed into potassium carbonate via potassium bicarbonate and formate intermediates¹. Moreover, the computed Gibbs free energies confirmed that the equilibrium between KHCO₃ and KOOCH is indeed feasible under our reaction conditions¹. Hence, we proposed a tandem reaction mechanism operated at standalone iron-phase: (i) While interacting with CO₂, Fe/K₂CO₃ phase could yield KCOOH and release CO, which is (ii) subsequently hydrogenated to olefins via conventional FTS route¹. Therefore, the primary role of KO₂ is to enhance the adsorption and activation of CO₂. On this note, we wish to refer to our previous publication for more discussion (including Figures 1 and 5 of ref. 14 in the revised manuscript).

2. As we known, Fe often shows paramagnetic character, which can raise the difficulty for the NMR detection of the species nearby. The analysis of the possible intermediates around the Fe-phase in the combined catalyst would be helpful to obtain comprehensive knowledge of the CO₂ conversion.

Response: We fully agree with this argument. Owing to the paramagnetic nature of the iron catalyst, it is indeed impossible to perform ssNMR experiments on the metallic phase. Moreover, direct (in-situ/operando) probing of iron-phase spectroscopically to detect carbonylates is also extremely challenging due to the technological and safety issues related to the reaction conditions. In our opinion, these challenges led to a lack of consensus on the FTS reaction mechanism, despite its (almost) one hundred years of history. In addition to the experimental evidence provided in ref. 14, in this work, we have additionally performed Raman (micro)spectroscopy and air-protected capillary single-crystal X-ray diffraction studies to demonstrate that no organic carbonylated species have been detected by us on the Fe-phase (Supplementary Fig. 18-20; also see Supplementary Section S2.1 for the detailed discussion).

Therefore, our research strategy to combine the iron-catalyst with zeolites gave us an interesting opportunity to probe the metallic phase *indirectly*. Since all products and intermediates of FTS stream would pass to the zeolite-phase in our dual-bed system, we designed this project to perform ssNMR experiments on the zeolite-phase after using fully ^{13}C -enriched CO_2 in the reactant feed. We understand and agree with the value and potential impact of this comment, and we will keep working in this direction to eliminate such technical obstacles. Herein, we refer to our newly written section on “Overall Reaction Process” (see Pages 16-23, including Fig. 7-8).

3. In Figure S3, why were the carbonyl species only detectable in the DE (direct excitation) experiments, but not in CP? In addition, the CP signal intensities of aromatics were weaker than in DE experiments. why?

Response: These observations are attributed to the respective mobility features of zeolite-trapped organics on the post-reacted materials. Herein, we would like to emphasize that we already have made a detailed discussion on both issues: (i) Page 9: Lines 7-14 in the manuscript and (ii) Supplementary Section S2.4 (cf. Page S19: Lines 8-22 and Page S20-21) in the supplementary information. To elucidate the reaction mechanism, we have employed diverse solid-state NMR magnetization transfer techniques to detect zeolite-trapped organic species based on their mobility. This strategy to utilize the ‘*mobility dependent*’ ssNMR technique allowed us to distinguish between mobile and rigid molecules on post-reacted zeolites.

Firstly, carbonyls were undetected in CP-based experiments due to their relatively mobile nature, since through-space (dipolar transfer such as in ^1H - ^{13}C CP) magnetization transfer schemes could only distinguish rigid/immobilized species (such as species physisorbed in/on the zeolite; e.g., aromatics). Contrary, direct excitation experiments *are meant to detect all* chemical species (rigid and mobile species, including those that exhibiting intermediate dynamics). Therefore, the carbonyl species were detectable in the DE only. Secondly, CP signal intensities of aromatics are indeed weaker (more specifically broader) than in DE experiments, which could be again attributed to their rigidity/restricted mobility nature. Since the CP detects the rigid species, by default, these rigid species have very short transverse (or spin-spin) relaxation time (T_2), which is directly linked to the broad linewidth (linewidth $\propto 1/T_2$), as also

observed in the present study. Therefore, aromatics signal intensities were weaker/broader in CP than in DE experiments. Herein, we refer to the supplementary information for a more detailed discussion (especially Page S19).

In essence, we also wish to highlight the significance of mobility and host-guest chemistry in zeolite catalysis through this work, which, we believe, deserves more research attention from the catalysis community.

4. Identification of exact species formed in a complicated reaction is not a trivial work. The authors attributed the ^{13}C signals at 197.5/23.05ppm to diacetyl on ZSM-5. Alternatively, these signals can also come from other species. Complementary experiments are expected to consolidate the assignments.

Response: We sincerely thank the reviewer for recognizing the challenge associated with our ssNMR-based investigation. We agree that an appropriate assignment of observed chemical shifts to a particular chemical structure is often not straightforward. During our research (not specifically to the present study), we always perform control chemisorption experiments, when we encounter any relatively unknown correlations. It means that the potential molecules are allowed to be adsorbed on zeolite and then check their individual spectral response for the verification. In this particular case of diacetyl, we have similarly validated our assignment: performing ssNMR experiments on diacetyl chemisorbed on zeolite ZSM-5 (Supplementary Fig. 12). Therefore, we are confident about our ‘diacetyl’ assignment in this study and its formation route has further been supported by theoretical calculation (cf. Figure 8 and Page 23 in the manuscript). Herein, we also wish to highlight that we already have made a detailed justification for our assignment protocols in the supplementary information (see Page S16: Lines 10-23 and Page S17: Lines 1-9 under Supplementary Section S2.4).

5. In figure 7, the authors proposed that carbonylation of $(\text{CH}_2)_n$ could generate ketene. How does this route operate? In addition, since the C-C hydrocarbons were formed, was it necessary to continue to proceed secondary reaction to form the so-called intermediate? It seems that carbonylation of methoxy species (generated by methanol) is a more reasonable route for the formation of initial C-C bond species according to the previous work (Angew. Chem. Int. Ed.

2006, 45, 1617-1620; J. Phys. Chem. C 2013, 117, 5840–5847; Angew. Chem. Int. Ed. 2016, 55, 5723–5726). In this mechanism, methyl acetate and acetyl were usually formed and the latter could be readily converted to ketone which was unstable (also not observed in this work) and could be further transformed to other species. Although methanol was solely observed by the authors on ZSM-5, the absence of methanol on other zeolites does not mean this route is not involved in the generation of ketone and related species.

Response: This is indeed a thought-provoking question. First, we agree that it is not mandatory to form ketene via carbonylation of $(\text{CH}_2)_n$. Also, we are confident that methanol is formed at the metallic phase, which is a ‘common factor’ applicable to all zeolites. Hence, we also agree that there is no exclusive relationship between methanol and ZSM-5, as the reviewer has also indicated. Since ketenes could also be formed from surface-acetate species, carbonylation of methoxy species (generated by methanol) could indeed be a more reasonable route for forming initial C-C bonds. We also agree with this assessment from the reviewer. Next, we also agree with the reviewer on inter-convertible relationships between acetyl, methyl acetate and ketone; we detected all three species in the present study.

Based on this comment from the reviewer (and also based on comments from other reviewers), we have now revised the mechanistic scheme completely (see Fig. 7-8). While designing this mechanistic scheme, we have also considered literature (in addition to our work) and expanded our theoretical calculations to provide further support. Based on our calculations, the ketene reaction with the Bronsted acid site leading to surface acetate is exothermic with reaction energies in the range of 31-47 kJ/mol, similar to previous calculations. (see ref. 46-47 in the revised manuscript)^{2,3}, with a negligible reaction barrier as well (<20 kJ/mol). In addition, previous calculations also showed that CO insertion could lead to surface acetate or even stable protonated ketene that can donate a proton to the active site and form neutral ketene with feasible energetics involved (Ref. 46,50 in the revised manuscript)^{2,4}. In this context, the new references proposed by the reviewer are included as well (Ref. 42,43,50 in the revised version). Finally, we wish to refer to the newly written sections on the full reaction mechanism and theoretical calculations (see Page 16-24, Table 1, Fig. 7-8) in the revised manuscript.

6. The authors proposed the C-C coupling of methoxyl with ketene led to the formation of

acetone. I am confused that how this route operates? Is hydrogen needed in this route? In addition, what is the meaning of “2*”?

Response: We thank the reviewer for the observation, and in response, we have corrected and detailed the reaction scheme accordingly. We refer to the “Theoretical Analysis of Zeolite Phase Chemistry” section in the revised manuscript, along with Table 1 and Fig. 8 (see Pages 21-24 in the revised manuscript). Also, in the previous version of the manuscript, “2*” meant two molecules of species. In the updated reaction scheme (Fig. 8), one methoxy group was used together with a hydrogenation step that led to a viable reaction route to acetone.

7. In this work, the authors introduced the “organic-inorganic” hybrid nature of the working zeolite, namely “supramolecular reactive center”, which had been proposed and investigated in the MTH reaction by previous work (J. Am. Chem. Soc. 2000, 122, 10726-10727; Angew. Chem. Int. Ed. 2016, 55, 2507-2511).

Response: This is indeed true. In the revised version, we made some minor modifications throughout the manuscript to accommodate the concept of ‘supramolecular reactive centers’. Also, we incorporated a few more references on this concept⁵⁻⁷, including two newly suggested references by the reviewer. Herein, we refer to the following sections: (i) the end of abstract, (ii) Page 5: Line 9-13 in the introduction section, (iii) the end of conclusion, (iv) ref. 34-35 in the revised manuscript.

Reviewer #2 (Remarks to the Author):

In this work, “Selectivity descriptors for the direct hydrogenation of CO₂ to hydrocarbons during zeolite-mediated bifunctional catalysis”, the authors provided a systematic study and carried out detailed characterization to understand the descriptors for the direct hydrogenation of CO₂ to hydrocarbons during zeolite-mediated bifunctional catalysis. There have been many papers published regarding metal oxide/zeolite composite catalysts to control the product selectivity including light olefins, branched hydrocarbons, and aromatics. In this study, the

authors used solid-state NMR spectroscopy and computational method to demonstrate the hybrid nature of the zeolite catalyst to produce different types of reaction intermediates for the ultimate product selectivity. It is an appreciable work for the catalysis research. However, the work does not have much more novelty to publish in Nature communications and the reasons are below.

Response: We thank reviewer 2 for the comments and suggestions. Our responses with respect to the specific queries are discussed below. We hope these discussions will help convince the reviewer of the challenging and innovative nature of our work and of the importance of the new mechanistic aspects here revealed.

1. The manuscript describes the influence of the zeolite framework, involving eight different zeolites, in combination with a Fe₂O₃@KO₂ catalyst and depending on the zeolite, various products such as light olefins (MOR, SAPO-34, ZSM-5, BETA, Y), paraffins (FER), long (olefinic) hydrocarbons (ZSM-22), and aromatics on (ZMS-5) were produced depending on the zeolite framework; it is well-known phenomena in the CO₂ hydrogenation. The authors are trying to rationalize the formation of intermediates/descriptors of the formation of hydrocarbons on Fe₂O₃@KO₂/zeolite. However, the authors are already published the data in their previous publication; ACS Catal. 2019, 9, 7, 6320–6334; using ZSM-5 and MOR with Fe₂O₃@KO₂. Solid-state nuclear magnetic resonance characterization of the zeolites revealed that the reaction mechanism is driven by the incorporation of CO in the network in the form of surface formate. Whereas, in this manuscript, the authors presented CO formed from the RWGS further converted to ketene/ketone/esters on the same Fe₂O₃@KO₂/zeolite. It is contradictory results.

Response: We understand the concern from the reviewer. *Firstly*, we wish to emphasize that there are no contradictory results presented in this paper compared to our earlier work⁸. We agree that the proposed reaction scheme in the previous version (Fig. 7) was presented in an overly simplistic way, which may have created this confusion. We regret this unintended issue, and therefore, we have illustrated a more detailed reaction scheme in the revised version (Fig. 7-8). This revised reaction scheme was proposed based on our experimental results as well as from the literature. Moreover, we have provided further theoretical and experimental support

to our proposal. Briefly, the primary source of CO is always RWGS during CO₂ hydrogenation (Fig. 7a). As we demonstrated before, CO can also be formed at the K-phase too (Fig. 7b). Afterward, traditional FTS chain initiation and chain propagation take place over the metal surface via the following two ways (Fig. 7c-d): (a) CO dissociative pathway (popularly known as carbide- or Biloen-Sachtler mechanism) and (b) CO insertion pathway (popularly known as Pichler-Schulz mechanism)⁹⁻¹³. Finally, in the zeolite phase, after surface methoxy species form on the zeolite active site, either via methanol reaction or hydrocarbon cracking, CO insertion follows, to result in zeolite bound acetyl that can further react with methanol to form methyl acetate. As presented in Fig. 8a, both the formation of surface acetate from methoxy ($E_r = -96$ kJ/mol) and methyl acetate from acetyl ($E_r = -95$ kJ/mol) are thermodynamically feasible, as previously reported^{3,4}. In the case of acetone formation, two reaction routes starting from ketene were proposed. Because of the high instability of the methylated ketene ($E_r = +56$ kJ/mol; Fig. 8b), an alternative route was investigated involving an initial hydrogenation of ketene to acetaldehyde ($E_r = 135$ kJ/mol) and further methylation ($E_r = -144$ kJ/mol; Fig. 8c) to acetone which occurred with a considerably high energy release. Additional carbonylation of acetone to diacetyl was also shown to be a viable reaction route ($E_r = -18$ kJ/mol; Fig. 8c). Since all intermediates are formed through exothermic reaction routes, the proposed mechanism is expected to be feasible and further contribute to the initiation of C₂₋₄ hydrocarbons production¹⁴⁻¹⁷.

Next, the existence of formate and acetate species *is not contradictory*⁸, rather indicative of the chain growth in FTS chemistry, via the CO-insertion mechanism⁹⁻¹². Although this mechanism was originally proposed a long time ago (ref. 52-55 in the revised manuscript)⁹⁻¹², the direct spectroscopic evidence was elusive until now. *Secondly*, we wish to highlight one technical aspect. In the previous manuscript, ssNMR was performed using naturally abundant CO₂ in the reactant feed, which has only 1.1% ¹³C. Although we have managed to get some preliminary information, we were still limited due to this ¹³C sensitivity issue. This limitation has prompted us to design the current project, where these post-reacted samples were prepared using fully enriched ¹³CO₂ in the reactant feed. It should be worth mentioning that such a high-pressure hydrogenation attempt on ¹³CO₂ has never been employed before. As a result, we are

now able to derive more information as we increased the sensitivity of the sample (>90%), which allowed us to detect ketene/ketone/esters on the same catalyst in this work. Hence, we emphasize again: *results reported in this work are not contradictory, rather completely complementary.*

To clear all ambiguities associated with our previous version of the manuscript, we have significantly rewritten both reaction mechanism and theoretical calculation sections in the revised manuscript (Page 16-23, Table 1, and Figure 7-8).

2. Identification of trapped molecules using 2D MAS solid-state NMR in the zeolite pores were previously reported in a number of literatures (e.g., ACS Catal. 2018, 8, 7356–7361). In addition, the formation of surface acetate, formate and methylated polyene/benzene in H-SAPO-34 was reported previously (Angew. Chem. Int. Ed. 2016, 55,15840-15845).

Response: Yes, it is indeed true. We already have included several such references in the previous version of the manuscript, including the Angewandte paper referred by the reviewer (ref. 28 in the manuscript). In the revised version, we have added a few more relevant publications, including the suggested ACS Catalysis paper (ref. 49 in the revised manuscript). Moreover, we would like to mention one striking difference: The analogous reaction intermediates were earlier detected on zeolite catalyzed MTH chemistry only, whereas we have detected conceptually similar intermediates during fundamentally different FTS chemistry, which has been elusive to date.

3. The authors describe CO is a co-catalyst in the reaction. We think that CO is not a co catalyst and it is the initiator for the hydrocarbon production. In page no 17, the authors reported that ^{13}CO predominantly was converted to rigid species. Then how can assume CO is a co-catalyst?

Response: Yes, we do absolutely agree. Herein, we have corrected our claim in the revised version of the manuscript. We have now described CO as “an initiator”, not as “a co-catalyst” for the production of hydrocarbons (See Page 14: Line 26 in the revised manuscript).

4. The reaction mechanism in page no 20 (Fig.7a); the role of $\text{Fe}_2\text{O}_3@ \text{KO}_2$ is only RWGS using MOR zeolite? It is difficult to believe! Because during the reaction Fe_2O_3 converted to Fe_5C_2 and C-C coupling reactions takes place on it?

Response: We again understand the concern. As we have responded to comment 1, this was not our original intention earlier. This issue has now been completely rectified, and we also have modified our reaction mechanism scheme. Therefore, to introduce more clarity, we now have provided a detailed mechanistic outlook as Fig. 7 in the current version of the manuscript, which we believe would clear all doubts from the reviewer's mind on this issue. Similar to comment 1, we refer to following newly written sections on the reaction mechanism and theoretical calculations in the revised manuscript (Page 16-23, Table 1, and Figure 7-8).

5. It is necessary to present ssNMR spectrum of the spent standalone Fe₂O₃@KO₂ catalyst and compare with ssNMR spectra of the Fe₂O₃@KO₂/zeolite. Based on recent study on in situ FT-IR of Fe-based catalysts, some of reaction intermediates are present in the Fe-based catalysts. Thus, the reaction intermediates that were discussed in the ssNMR spectra of the Fe₂O₃@KO₂/zeolite may not solely from the zeolite.

Response: We believe that there is some confusion again. First of all, we wish to emphasize that it is impossible to perform ssNMR experiments on the spent standalone iron catalyst due to the paramagnetic nature of metallic catalyst (as also acknowledged by reviewer 1). Secondly, we never claimed in the manuscript that all reaction intermediates were “solely” derived from/by zeolites. As we always emphasized throughout the manuscript, the metallic phase is responsible for the CO₂ hydrogenation, and zeolite is ‘controlling’ the final product distribution (see 3rd paragraph in our introduction). In our dual-bed system, since all FTS-based products and intermediates would pass to the zeolite phase, we designed this project to perform ssNMR experiments on the zeolite phase to probe the reaction mechanism. However, based on this comment, we have also elaborated our discussion on the mechanism section to provide more clarification and eliminate doubts. Moreover, we have provided two new mechanistic Figures (Fig. 7-8), where we have distinctly illustrated the individual role of metallic and zeolite phases. Also, the reaction pathways on the zeolite-phase have further been investigated by the computational analysis in this work. For a detailed discussion, therefore, we again refer to the following sections: Page 16-23, Table 1, and Figure 7-8.

6. The authors presented the reaction mechanism for the formation of methanol in Fig. 7b. However, there is no reaction data or ssNMR spectrum to prove the presence of methanol in

either the manuscript or in the supplementary data. How do you confirm the addition of methanol in the ketene on the zeolite?

Response: Firstly, we must highlight that we already have provided sufficient spectroscopic evidence of methanol and dimethyl ether in the previous version of the manuscript. For their evidence, we refer to the following sections: (i) Figure 5f in the manuscript, as well as related discussion in (ii) Page 13: Line 5-7 in the manuscript and (iii) more detailed section 2.4 in the Supplementary Information.

With respect to the second query related to the addition of methanol in the ketene on zeolites, previous calculations (e.g.: ref. 47 in the revised manuscript) analyzing methanol reacting with ketene determined a significantly high free energy reaction barrier (-143 kJ/mol) to forming surface propionate species, although the Gibbs reaction energy was exothermic (61 kJ/mol)³. To respond to the reviewer's suggestion, further calculations of propionate formation on the four high adsorbing zeolites were conducted, which found that the reaction energies fall in the 108-113 kJ/mol interval. However, it would be worth noting that methanol can be easily consumed in several ways, either the typical methanol-to-hydrocarbons route or CO-related process such as methanol reacting with surface-acetate to form methyl acetate, which is more energetically feasible than methanol ketene reaction (see ref. 47 in the revised manuscript)³.

7. Most of the ssNMR spectra are the trapped molecules in the spent zeolite. Since these compounds are physically adsorbed, and not further converted during the reaction, how can say these are descriptors for the CO₂ hydrogenation reaction? Really these species are descriptors or spectators?

Response: As we already have highlighted in the manuscript and the reaction mechanism, all these observed species were independently capable enough to produce hydrocarbons on the zeolite¹⁷⁻²¹. Owing to this feature, supramolecular reactive centers (constituted by inorganic zeolites and organic hydrocarbons)⁵⁻⁷ are acknowledged as an active or working catalyst during zeolite-catalyzed hydrocarbon conversion. It is worth mentioning that reviewer 1 has also acknowledged this fact. Herein, we would refer to the following articles in the reference list, where the 'descriptor' nature of observed organic hydrocarbons and carbonylated species has

clearly been demonstrated, primarily by the groups of Haw, Deng/Xu, Svelle, Weckhuysen, and us (ref. 27, 31-36 in the revised manuscript)^{5-7,17-20}. Moreover, a similar product distribution during our control experiments also supports the impact of HCP-based reaction centers within the zeolite. Next, not all observed species were physically adsorbed, as we have highlighted their mobility feature, that too individually, throughout the manuscript. Therefore, we can certainly conclude that these trapped molecules are not spectators. Furthermore, in the revised manuscript, we have included an additional discussion highlighting the feasibility of certain carbonylated species to interconvert between each other based on the calculated reaction energies (see Fig. 7-8 and Table 1 in the manuscript), as also previously responded at the first query of the second reviewer. Collectively, we wish to refer to the following sections of the manuscript for more in-depth discussion: Page 16-23, Table 1, and Figure 7-8. Additionally, we also refer to our response to reviewer 3 on this matter.

8. In the computation study, full reaction pathways with activation energies, comparison of reaction coordinates of possible reaction pathways should be presented.

Response: In this work, our primary aim was to identify the influential reaction intermediate and their potential formation routes. However, in principle, it is completely viable that one particular intermediate could form via multiple routes. This task becomes more complicated when different zeolites with diverse characteristics are involved. To provide more reliable computational analysis, we should explore all possible routes on all zeolites. As the reviewer can imagine, this a very time-consuming task, which should be a separate project or thesis work by itself. It means that providing full energetics and relative comparison of reaction coordinates of possible reaction pathways are a non-pragmatic option for us in this work. Therefore, in this work, we have determined the reaction energies involved only for one case to clarify the thermodynamics involved in order to have a proper perspective on the feasibility of the aforementioned reaction paths, which are further detailed in Section “Theoretical analysis of zeolite phase chemistry” at Page 21-23 (also, Figure 8, Table 1 in the manuscript and Supplementary Section S2.5 in the Supplementary Information). From our side, we commit that we will be continuing to work on expanding that analysis by determining the reaction barriers involved and expand the set of potential reaction paths available.

Reviewer #3 (Remarks to the Author):

The authors synthesized the Fe-based catalysts for CO₂ hydrogenation to olefins, aromatics and paraffins. It is interesting that the reaction intermediates in different zeolites were characterized. Different zeolite and reaction intermediates were found to govern the ultimate product selectivity. However, some questions need to be addressed before its publication, the details are as follows:

Response: We thank reviewer 3 for the encouraging comments, suggestions, and positive evaluation. Our responses with respect to the specific queries are discussed below.

1. The title of this manuscript is “selectivity descriptors for the direct hydrogenation of CO₂ to ...” , however the detail “descriptors” for the selectivity of CO₂ hydrogenation was not clearly described.

Response: We understand now that there was a ‘knowledge gap’ in the previous version of the manuscript. We originally intended to describe the ‘organic-inorganic’ hybrid nature of zeolites (or popularly known as supramolecular reaction centers) as the ‘descriptor’ too. We did not intend to highlight topology alone as a descriptor, as it might be non-reasonable to do so (also indicated by reviewer 3 in the last comment). The unique relationship between the inorganic zeolite and organic hydrocarbon pool-based reaction centers is crucial while controlling the ultimate selectivity (See Fig. 7e-f in the revised manuscript for a simplified illustration). While responding to comment no. 7 from reviewer 2, we also have highlighted pioneering work on this concept, mostly in the field of methanol-to-hydrocarbon (ref. 27, 31-36 in the revised manuscript)^{5-7,17-20}. Herein this work, interestingly, we have discovered several mechanistic resemblances, which prompted us to make such ‘descriptors’ claim. To introduce more clarity in our argument, we have now elaborated and rewritten the complete mechanistic discussion and computational analysis (Pages 16-23 in the manuscript) sections as well as modified the abstract and conclusion appropriately in the revised manuscript. Moreover, we also have

provided additional explanations while responding to the last comment from the reviewer (see below).

2. The manuscript gave the detailed information for the reaction intermediates in different zeolites by ssNMR and the calculation. I think the reaction intermediates and product were well characterized. However, it only gave a phenomena, but not the intrinsic reason.

Response: Yes, we absolutely agree. In the quest of finding the intrinsic reason, the appropriate design of experiments is crucial, which demands more time and effort. In this work, our original intention was to provide direct spectroscopic evidence for reactive intermediates in this catalysis, which has not been elusive till now. We will be continuing to work in this direction to unravel more insights about the reaction mechanism.

3. The eight different zeolite topologies were classified into four groups based on the selectivity, so the topology should be the descriptor, then why the zeolites with the same topology showed different selectivity?

Response: As we admitted before, it was not our intention to highlight topology alone as a descriptor, but rather to identify the main factors governing product selectivity. In addition to the zeolite topologies, the hybrid ‘organic-inorganic’ supramolecular reaction centers are descriptors as well and collectively govern the ultimate product selectivity (similar to MTH chemistry) (See our newly drawn Fig. 7e-f in the revised manuscript). Therefore, we have significantly rewritten both reaction mechanisms and theoretical calculation sections (Pages 16-23) to introduce more clarity and avoid further confusion. Additionally, we have made appropriate modifications in the abstract, introduction, and conclusion sections.

Herein, we wish to summarize the arguments made and studies referenced throughout the main manuscript, highlighting that the relationship between selectivity and topology is a complicated task, as also indicated by the reviewer in this comment. Due to the molecular size of zeolite pores and the numerous topologies available, the structure of the hydrocarbons can be “molded” to a specific type. On this matter, we specifically refer to our mechanistic subsection on zeolite (Pages 18-19 and ref. 58-60 in the revised manuscript)²²⁻²⁴. Therein, we refer to a comprehensive study performed by Olsbye et al., at 400°C and 80% methanol

conversion on numerous 1D large pore zeolites, including ZSM-22 (TON with 10-ring elliptical channel) and ZSM-23 (MTT with 10-ring teardrop channel), which could deliver C₅₊-aliphatics products without any aromatics. However, ZSM-5 (MFI 3D 10-ring channel with cross-sections) is more appropriate for producing aromatics among 10-member-ring zeolites, as also demonstrated in the current study. Furthermore, the structure of the individual pores forming the zeolite and the overall zeolite topology are needed to consider while explaining the product shape selectivity. As Weckhuysen et al. reported in ref.30 in the revised manuscript²⁵, the sinusoidal channel of H-ZSM-5 favors the olefin/paraffin cycle, while the straight channel facilitates aromatics production. Moreover, based on the theoretical calculations conducted in a wide range of hydrocarbons, we should also consider the diffusion rate of hydrocarbons through zeolite pores, which is dependent on the channel's height, width, and shape (ref. 61-62 in the revised manuscript)^{26,27}. It essentially means that the molecular fit and selectivity of product outflow, is dependent on the structural inhibition imposed by the molecular size and shape of the zeolite pores. Although the zeolite lattice ensures the necessary space to form a specific hydrocarbon, zeolites with identical topologies reported performing differently due to an uneven acid site strength [ref. 58,65,66 in the revised manuscript]^{22,28,29}. Besides acid strength, acid site density also plays a critical role in influencing the shape and reactivity of the "hydrocarbons pool" and hence, governing the overall product distribution [ref. 58,67,68 in the revised manuscript]^{22,30,31}.

As also illustrated in Fig. 7e-f, both zeolite topologies and 'supramolecular reaction centers' (i.e., hydrocarbons' or 'hydrocarbons pool'- adsorbed and confined by the inorganic part -'acid site and surrounding zeolite lattice') are governing the ultimate product selectivity, and thus, are considered as *descriptors* during the catalytic processes that take place inside the zeolite pores. Therefore, we have now modified our discussion throughout the manuscript, including abstract, introduction, and conclusion sections to clarify and distinguish the role of zeolite topology and 'supramolecular reaction centers' on product selectivity.

Bibliography

1. Ramirez, A. *et al.* Tandem conversion of CO₂ to valuable hydrocarbons in highly concentrated potassium iron catalysts. *ChemCatChem* **11**, 2879–2886 (2019).
2. Rasmussen, D. B. *et al.* Ketene as a reaction intermediate in the carbonylation of dimethyl ether to methyl acetate over mordenite. *Angew. Chem. Int. Ed.* **54**, 7261–7264 (2015).
3. Plessow, P. N. & Studt, F. Unraveling the mechanism of the initiation reaction of the methanol to olefins process using ab Initio and DFT calculations. *ACS Catal.* **7**, 7987–7994 (2017).
4. Liu, Y. *et al.* Formation Mechanism of the First Carbon-Carbon Bond and the First Olefin in the Methanol Conversion into Hydrocarbons. *Angew. Chemie Int. Ed.* **55**, 5723–5726 (2016).
5. Song, W., Haw, J. F., Nicholas, J. B. & Heneghan, C. S. Methylbenzenes are the organic reaction centers for methanol-to-olefin catalysis on HSAPO-34 [12]. *Journal of the American Chemical Society* vol. 122 10726–10727 (2000).
6. Wang, C. *et al.* Direct Detection of Supramolecular Reaction Centers in the Methanol-to-Olefins Conversion over Zeolite H-ZSM-5 by ¹³C-27Al Solid-State NMR Spectroscopy. *Angew. Chemie - Int. Ed.* **55**, 2507–2511 (2016).
7. Haw, J. F. & Marcus, D. M. Well-defined (supra)molecular structures in zeolite methanol-to-olefin catalysis. *Top. Catal.* **34**, 41–48 (2005).
8. Ramirez, A. *et al.* Effect of Zeolite Topology and Reactor Configuration on the Direct Conversion of CO₂ to Light Olefins and Aromatics. *ACS Catal.* **9**, (2019).
9. Hall, W. K., Kokes, R. J. & Emmett, P. H. Mechanism Studies of the Fischer-Tropsch Synthesis. The Addition of Radioactive Methanol, Carbon Dioxide and Gaseous Formaldehyde. *J. Am. Chem. Soc.* **79**, 2983–2989 (1957).
10. Davis, B. H. Fischer-Tropsch Synthesis: Reaction mechanisms for iron catalysts. *Catal. Today* **141**, 25–33 (2009).

11. Biloen, P., Helle, J. N. & Sachtler, W. M. H. Incorporation of surface carbon into hydrocarbons during Fischer-Tropsch synthesis: Mechanistic implications. *J. Catal.* **58**, 95–107 (1979).
12. Pichler, H. & Schulz, H. Neuere Erkenntnisse auf dem Gebiet der Synthese von Kohlenwasserstoffen aus CO und H₂. *Chemie Ing. Tech.* **42**, 1162–1174 (1970).
13. Blyholder, G. & Emmett, P. H. Fischer-Tropsch synthesis mechanism studies. The addition of radioactive ketene to the synthesis gas. *J. Phys. Chem.* **63**, 962–965 (1959).
14. Song, W., Nicholas, J. B. & Haw, J. F. A persistent carbenium ion on the methanol-to-olefin catalyst HSAPO-34: Acetone shows the way. *J. Phys. Chem. B* **105**, 4317–4323 (2001).
15. Xu, T., Haw, J. F. & Munson, E. J. Toward a Systematic Chemistry of Organic Reactions in Zeolites: In situ NMR Studies of Ketones. *J. Am. Chem. Soc.* **116**, 1962–1972 (1994).
16. Vogt, C., Weckhuysen, B. M. & Ruiz-Martínez, J. Effect of Feedstock and Catalyst Impurities on the Methanol-to-Olefin Reaction over H-SAPO-34. *ChemCatChem* **9**, 183–194 (2017).
17. Chowdhury, A. D. *et al.* Bridging the gap between the direct and hydrocarbon pool mechanisms of the methanol-to-hydrocarbons process. *Angew. Chem. Int. Ed.* **57**, 8095–8099 (2018).
18. Yarulina, I., Chowdhury, A. D., Meirer, F., Weckhuysen, B. M. & Gascon, J. Recent trends and fundamental insights in the methanol-to-hydrocarbons process. *Nat. Catal.* **1**, 398–411 (2018).
19. Svelle, S. *et al.* Conversion of methanol into hydrocarbons over zeolite H-ZSM-5: Ethene formation is mechanistically separated from the formation of higher alkenes. *J. Am. Chem. Soc.* **128**, 14770–14771 (2006).
20. Bailleul, S. *et al.* A Supramolecular View on the Cooperative Role of Brønsted and Lewis Acid Sites in Zeolites for Methanol Conversion. *J. Am. Chem. Soc.* **141**, 14823–14842 (2019).

21. Chowdhury, A. D. *et al.* Initial carbon-carbon bond formation during the early stages of the methanol-to-olefin process proven by zeolite-trapped acetate and methyl acetate. *Angew. Chem. Int. Ed.* **55**, 15840–15845 (2016).
22. Olsbye, U. *et al.* Conversion of Methanol to Hydrocarbons: How Zeolite Cavity and Pore Size Controls Product Selectivity. *Angew. Chem. Int. Ed.* **51**, 5810–5831 (2012).
23. Rojo-Gama, D. *et al.* Time- and space-resolved study of the methanol to hydrocarbons (MTH) reaction-influence of zeolite topology on axial deactivation patterns. *Faraday Discuss.* **197**, 421–446 (2017).
24. Etemadi, S. Catalytic investigations of zeolite based methanol to hydrocarbons catalysts. (2015).
25. Fu, D. *et al.* Elucidating zeolite channel geometry–reaction intermediate relationships for the methanol-to-hydrocarbon process. *Angew. Chem. Int. Ed.* **59**, 20024–20030 (2020).
26. Smit, B. & Maesen, T. L. M. Towards a molecular understanding of shape selectivity. *Nature* vol. 451 671–678 (2008).
27. Van Speybroeck, V. *et al.* Advances in theory and their application within the field of zeolite chemistry. *Chemical Society Reviews* vol. 44 7044–7111 (2015).
28. Bleken, F. *et al.* The effect of acid strength on the conversion of methanol to olefins over acidic microporous catalysts with the CHA topology. *Top. Catal.* **52**, 218–228 (2009).
29. Knott, B. C. *et al.* Consideration of the aluminum distribution in zeolites in theoretical and experimental catalysis research. *ACS Catal.* **8**, 770–784 (2018).
30. Di Iorio, J. R., Nimlos, C. T. & Gounder, R. Introducing catalytic diversity into single-site chabazite zeolites of fixed composition via synthetic control of active site proximity. *ACS Catal.* **7**, 6663–6674 (2017).
31. Nastase, S. A. F. *et al.* Mechanistic Insight into the Framework Methylation of H-ZSM-5 for Varying Methanol Loadings and Si/Al Ratios Using First-Principles Molecular Dynamics Simulations. *ACS Catal.* **10**, 8904–8915 (2020).

REVIEWERS' COMMENTS

Reviewer #1 (Remarks to the Author):

I am satisfied with the response and the modifications made by the authors, and recommend to accept the revised manuscript for publication in Nature Communication as is

Reviewer #3 (Remarks to the Author):

The authors have addressed my comments from the original manuscript. I recommend for its publication.

Point-by-Point Response to Reviewers' Comments

Reviewers' comments are written in **blue**

Authors' responses are in **black**

Reviewer #1 (Remarks to the Author):

I am satisfied with the response and the modifications made by the authors, and recommend to accept the revised manuscript for publication in Nature Communication as is.

Response: We thank reviewer 1 for the encouraging comments, suggestions, and positive recommendations during the evaluation of our manuscript.

Reviewer #3 (Remarks to the Author):

The authors have addressed my comments from the original manuscript. I recommend for its publication.

Response: We thank reviewer 3 for the encouraging comments, suggestions, and positive recommendations during the evaluation of our manuscript.